# Bidirectional promoter activity from expression cassettes can drive off-target repression of neighboring gene translation

Emily Nicole Powers[1], Charlene Chan[1], Ella Doron-Mandel[2], Lidia Llacsahuanga Allcca[1], Jenny Kim Kim[2], Marko Jovanovic[2], Gloria Ann Brar[1,3,4]*

[1]Department of Molecular and Cell Biology, University of California, Berkeley, Berkeley, United States; [2]Department of Biological Sciences, Columbia University, New York, United States; [3]California Institute for Quantitative Biosciences (QB3), University of California, Berkley, Berkley, United States; [4]Center for Computational Biology, University of California, Berkeley, Berkeley, United States

*For correspondence: gabrar@berkeley.edu

**Competing interest:** The authors declare that no competing interests exist.

**Abstract** Targeted selection-based genome-editing approaches have enabled many fundamental discoveries and are used routinely with high precision. We found, however, that replacement of *DBP1* with a common selection cassette in budding yeast led to reduced expression and function for the adjacent gene, *MRP51*, despite all *MRP51* coding and regulatory sequences remaining intact. Cassette-induced repression of MRP51 drove all mutant phenotypes detected in cells deleted for *DBP1*. This behavior resembled the 'neighboring gene effect' (NGE), a phenomenon of unknown mechanism whereby cassette insertion at one locus reduces the expression of a neighboring gene. Here, we leveraged strong off-target mutant phenotypes resulting from cassette replacement of *DBP1* to provide mechanistic insight into the NGE. We found that the inherent bidirectionality of promoters, including those in expression cassettes, drives a divergent transcript that represses *MRP51* through combined transcriptional interference and translational repression mediated by production of a long undecoded transcript isoform (LUTI). Divergent transcript production driving this off-target effect is general to yeast expression cassettes and occurs ubiquitously with insertion. Despite this, off-target effects are often naturally prevented by local sequence features, such as those that terminate divergent transcripts between the site of cassette insertion and the neighboring gene. Thus, cassette-induced off-target effects can be eliminated by the insertion of transcription terminator sequences into the cassette, flanking the promoter. Because the driving features of this off-target effect are broadly conserved, our study suggests it should be considered in the design and interpretation of experiments using integrated expression cassettes in other eukaryotic systems, including human cells.

## Editor's evaluation

The power of yeast genetics frequently depends on the insertion of selectable expression cassettes. The authors demonstrate that an unfortunate vulnerability of these cassettes lies in the inevitable divergent antisense transcription that is produced, which can suppress the expression of proximal genes. The authors provide mechanistic insight into this consequence of yeast genomic editing and provide solutions that can be used for all such cassettes.

## Introduction

Genome engineering to create targeted gene deletions, mutations, reporter constructs, and epitope-tagged proteins is a key strategy for the mechanistic dissection of almost any biological process. Budding yeast was the first eukaryotic organism in which this became highly facile, thanks to the development of a one-step PCR-based editing strategy, frequently used with a shared toolkit of selection cassettes (*Longtine et al., 1998*; *Wach et al., 1997*; *Wach et al., 1994*; *Baudin et al., 1993*; *Lorenz et al., 1995*). These tools became common, routinely used by thousands of labs as well as to enable large global endeavors, such as creation of the yeast deletion collection (*Giaever et al., 2002*). Because of the high fidelity, ease of use, and rapid nature of this selection-mediated strategy, it has remained commonplace despite the development of methods allowing non-selection-marked mutations that include CRISPR/Cas9-based editing (*Jinek et al., 2012*; *Güldener et al., 1996*; *Gray et al., 2005*).

While selection-mediated editing strategies have been used to advance countless discoveries, their utility relies on the assumption that insertion of a selection cassette at one locus will not disrupt the expression of neighboring genes. However, analyses of mutant phenotypes detected in global studies of the collection of strains in which each non-essential yeast ORF is replaced with a kanMX cassette revealed effects that appeared to result from neighboring gene mis-regulation, termed 'the neighboring gene effect' (NGE) (*Ben-Shitrit et al., 2012*; *Atias et al., 2016*; *Egorov et al., 2021*). This is caused by overlap of the deleted ORF with a regulatory region for the neighboring gene in a subset of cases, as just one of several problems that resulted from the large-scale nature of the effort required to create the deletion collection (*Ben-Shitrit et al., 2012*; *Hughes et al., 2000*; *Teng et al., 2013*). However, most cases of the NGE remain unexplained, and this lack of mechanistic understanding makes it unclear how prevalent this effect is in traditional small-scale laboratory studies, and how to prevent it.

It is now understood that genomic loci can have complex and linked transcriptional outputs, even in yeast. For example, activity from one transcription start site (TSS) can interfere with the output of nearby TSSs in an adjacent sense or antisense configuration, and that most, if not all, promoters are bidirectionally active (*Teodorovic et al., 2007*; *Neil et al., 2009*; *Preker et al., 2008*; *Core et al., 2008*; *Xu et al., 2009*; *Seila et al., 2008*; *Churchman and Weissman, 2011*; *Jin et al., 2017*). Transcription interference is seen in diverse organisms, including yeast and human cells, and can occur between TSSs driving coding or non-coding RNAs and controlling both transcript isoform identity and transcript levels (*Chia et al., 2017*; *Hirschman et al., 1988*; *Martens et al., 2004*; *van Werven et al., 2012*; *Hongay et al., 2006*; *Hausler and Somerville, 1979*; *Adhya and Gottesman, 1982*; *Proudfoot, 1986*; *Boussadia et al., 1997*; *Emerman and Temin, 1984*; *Corbin and Maniatis, 1989*; *Struhl, 1985*). It has also been shown that more than one TSS is often present at a single locus, even in the simple budding yeast (*Pelechano et al., 2013*).

Transcription interference between TSSs within the same genomic locus is an important feature of a naturally occurring type of regulation dependent on long undecoded transcript isoforms (LUTIs; *Chia et al., 2017*; *Chen et al., 2017*). For genes regulated by this strategy, 5′ extended LUTIs are transcribed in place of canonical mRNAs to temporally downregulate protein synthesis (*Chia et al., 2017*; *Chen et al., 2017*; *Cheng et al., 2018*; *Van Dalfsen et al., 2018*). Here, use of an upstream alternate TSS represses transcription from the canonical TSS in cis via transcriptional interference (*Chia et al., 2017*). In concert, translation of the main ORF-encoded protein products from LUTIs are repressed by translation of competitive AUG-initiated upstream ORFs (uORFs) (*Chen et al., 2017*; *Wethmar, 2014*; *Barbosa et al., 2013*; *Hinnebusch et al., 2016*; *Law et al., 2005*; *Brar et al., 2012*; *Cheng et al., 2018*). Natural LUTI-based regulation has been shown to modulate gene expression for many genes during meiosis and the unfolded protein response in yeast (*Chen et al., 2017*; *Cheng et al., 2018*; *Van Dalfsen et al., 2018*). Furthermore, features of LUTI-based regulation are highly conserved across eukaryotes, and evidence suggests the presence of this regulation in more complex species, such as humans and algae (*Hollerer et al., 2019*; *Sehgal et al., 2008*; *Moseley et al., 2002*). Here, we show that insertion of expression cassettes commonly used edit the genomes of yeast cells, can induce the repression of neighboring genes though synthetic and constitutive LUTI-based repression.

In particular, we report that selection-cassette replacement of the ORF for *DBP1* causes mis-regulation of the adjacent gene, *MRP51*, despite no changes to the *MRP51* coding or regulatory regions, as a result of synthetic LUTI-based repression. All phenotypes we observed in cells with a

cassette-mediated deletion of the *DBP1* ORF were caused by mis-regulation of *MRP51*, rather than loss of *DBP1*, consistent with an NGE. We leveraged the strong off-target effects observed with cassette-mediated replacement of *DBP1* to interrogate the mechanism driving this phenomenon. We find that cassette-mediated off-target effects were general to all expression cassettes that we inserted at the *DBP1* locus and are driven by the bidirectional cassette promoter activity now understood to be inherent to eukaryotic promoters.

These data point to an undesirable feature of expression cassette insertion that is currently being overlooked in the design of genome-editing approaches. While we find that stable cassette-driven divergent transcripts were detected in ~30% of cassette-inserted loci, all cassette-inserted loci have detectable divergent transcripts in cells lacking nuclear exosome-mediated RNA decay. Thus, our data suggest that divergent transcription results ubiquitously from expression cassette insertion, but features—including transcription termination sequences adjacent to cassette insertion—can naturally mitigate off-target neighboring gene mis-regulation. Because it is difficult to predict from sequence information alone whether a locus will be sensitive to neighboring gene mis-regulation, we designed improved expression cassettes containing an additional terminator sequence flanking the promoter, which prevents neighboring gene disruption. Use of such cassettes should improve the specificity of engineered mutant strains for future studies. More broadly, our study uncovers a mechanism by which genome engineering can drive neighboring gene mis-expression, and that warrants consideration in the design of future studies in yeast, as well as other eukaryotes.

## Results

### Insertion of a resistance cassette at the *DBP1* locus causes aberrant transcription and reduced protein production from *MRP51*, a neighboring gene

We had previously observed upregulation of the Dbp1 RNA helicase in meiotic yeast cells, coincident with downregulation of its paralog, Ded1 (*Brar et al., 2012*). Ded1 is important for translation initiation, and has been well studied under conditions of mitotic exponential growth (*de la Cruz et al., 1997*). In contrast, the role of Dbp1 has remained less clear, partially a result of its absent or low expression under commonly studied laboratory conditions. Based on the identity of Dbp1 as an RNA helicase (*Jamieson and Beggs, 1991*), we predicted that Dbp1 could have a role in regulating gene expression during meiosis. To test this, we deleted *DBP1* by replacing the ORF (from start to stop codon) with a standard cassette-encoding Geneticin (G418) resistance, amplified from the pFA6a-kanMX6 plasmid (*Longtine et al., 1998*), in the SK1 budding yeast strain background (*Padmore et al., 1991*). We then compared gene expression profiles of wild-type and *dbp1Δ::kanMX6* cells during meiosis, using ribosome profiling and mRNA-sequencing (mRNA-seq). To our surprise, *MRP51*, the gene located directly adjacent to *DBP1*, exhibited a profound decrease in translation in the *dbp1Δ::kanMX6* strain relative to wild-type controls (*Figure 1A*). Translation of *MRP51* was 8.4-fold lower in *dbp1Δ::kanMX6* cells despite a 1.8-fold increase in *MRP51* mRNA abundance (*Figure 1A,B*). These changes reflected a 15-fold decrease in *MRP51* translation efficiency (TE) in the *dbp1Δ::kanMX6* cells compared to the wild-type control (*Figure 1C*; TE: ribosome footprint RPKM/mRNA RPKM) and led to a reduction in Mrp51 protein level compared to wild-type, as assessed by western blotting (*Figure 1D*). Because of the close genomic proximity of these genes, we hypothesized that this change in TE could be due to cis- effects from the *dbp1Δ::kanMX6* insertion rather than reflecting regulation of *MRP51* by Dbp1 protein. Upon closer examination of the mRNA transcripts produced at this locus, we found that replacing the *DBP1* ORF with the kanMX6 resistance cassette led to production of a 5′ extended *MRP51* mRNA compared to that in wild-type cells. The extended transcript contained 3 AUG-initiated uORFs not present on the wild-type transcript, which were translated at the expense of the *MRP51* ORF in *dbp1Δ::kanMX6* cells (*Figure 1E*).

These data suggest that kanMX6 cassette insertion at the *DBP1* locus drives the transcription of an aberrant 5′ extended *MRP51* mRNA containing repressive uORFs and causes reduced Mrp51 protein production. Thus, the features of *MRP51* mis-regulation in this strain mimicked the natural LUTI-based mode of regulation that conditionally downregulates protein synthesis from many genes in yeast (*Figure 1F*; *Chia et al., 2017*; *Chen et al., 2017*; *Cheng et al., 2018*; *Van Dalfsen et al., 2018*). Here, the primary difference was that repression of *MRP51* expression appeared to occur constitutively as

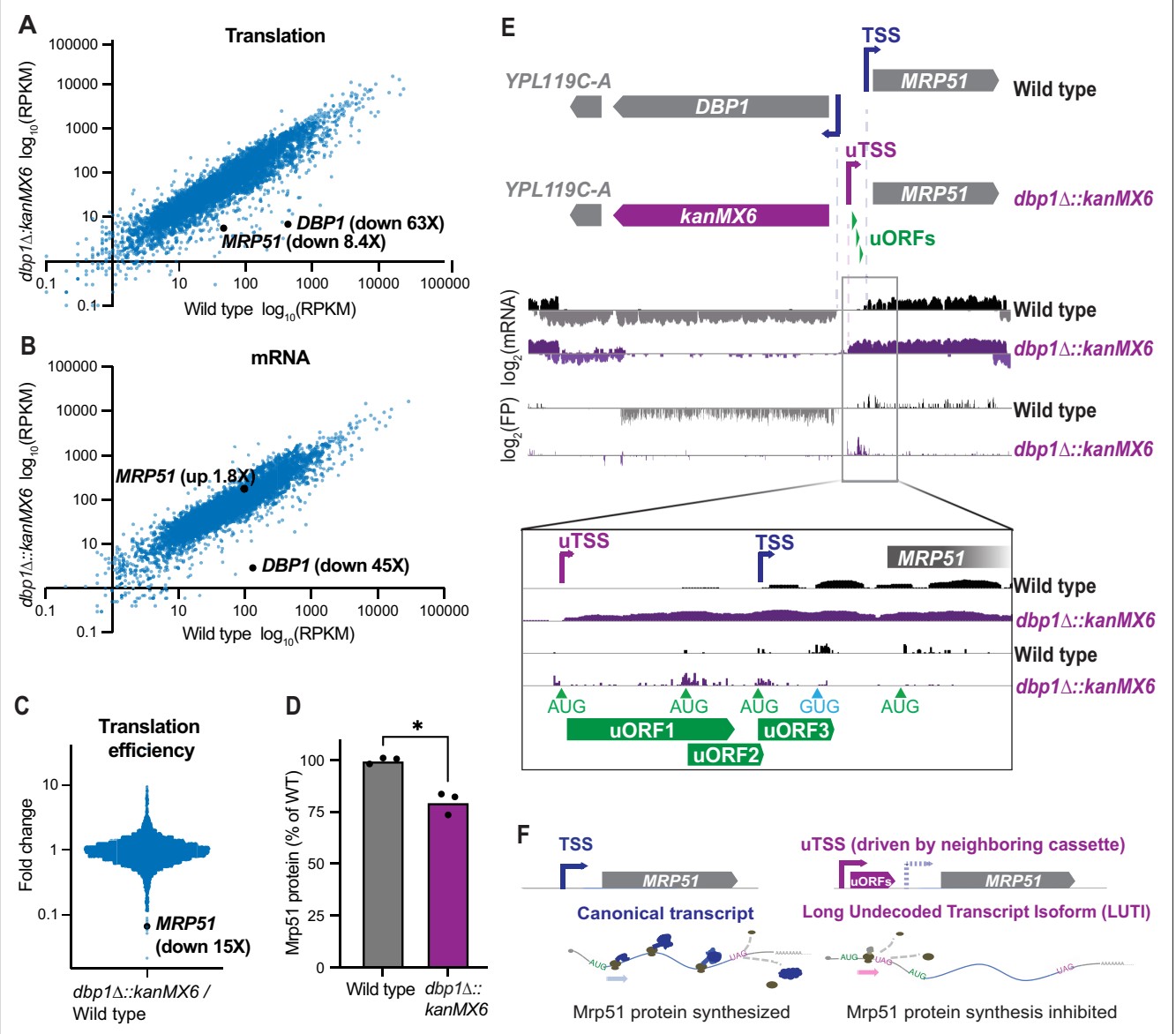

**Figure 1.** Insertion of a resistance cassette at the *DBP1* locus causes aberrant transcription and reduced protein production from the neighboring *MRP51* gene. (**A**) Translation (ribosome profiling, footprint) and (**B**) mRNA abundance (mRNA-seq) reads per kilobase million mapped reads (RPKM) for every ORF expressed in wild-type and *dbp1Δ::kanMX6* cells is plotted. (**C**) Fold-change of translation efficiency (TE: FP RPKM/mRNA RPKM) for all expressed genes. (**A–C**) Data represent RPKM values from a single experiment, for RPKM values for all quantified genes, see *Figure 1—source data 1*. (**D**) Quantification of Mrp51 levels in wild-type and *dbp1Δ::kanMX6* cells undergoing mitotic growth as determined by western blotting. Mrp51 levels were normalized to alpha tubulin and three independent biological replicates were quantified. Statistical significance was determined by a ratio paired *t*-test with a reported two-tailed p < 0.05. For representative blot see *Figure 1—figure supplement 1A*. (**E**) mRNA and FP reads mapped to the *DBP1/MRP51* locus in wild-type (gray/black) and *dbp1Δ::kanMX6* (purple) cells. Note that *MRP51* transcripts are 5' extended in the *dbp1Δ::kanMX6* cells compared to wild-type (see inset mRNA tracks). The extended transcript contains three AUG-initiated upstream ORFs (uORFs) translated at the expense of the *MRP51* ORF (see FP tracks and inset). (**F**) Model: replacement of *DBP1* ORF with a resistance cassette causes aberrant expression of a long undecoded transcript isoform (LUTI) for *MRP51*, which results in lower Mrp51 protein expression as an off-target effect.

The online version of this article includes the following source data and figure supplement(s) for figure 1:

**Source data 1.** RPKM values for mRNA-seq and ribosome profiling data for all genes quantified in the experiment shown in *Figure 1* plots.

**Figure supplement 1.** A published dataset confirms aberrant transcription and mis-regulation of *MRP51* following cassette-mediated *DBP1* replacement.

**Figure supplement 1—source data 1.** Zipped folder containing uncropped tif image of western blot shown in *Figure 1* with and without labels on the relevant samples and bands.

result of kanMX6 insertion at the neighboring *DBP1* ORF, rather than resulting from temporally regulated toggling between two TSSs.

We were surprised to observe this dramatic off-target mis-regulation because of the widespread and long-term use of cassette-mediated gene deletion and the detailed quality control measures that were used in construction of our strains to prevent off-target effects caused by background mutations or improper cassette insertion. To confirm aberrant *MRP51* regulation was not an artifact of our strain, experimental conditions, or selection cassette, we analyzed a published ribosome profiling and mRNA-seq dataset comparing wild-type and *dbp1Δ* cells of the S288C budding yeast background. These *dbp1Δ* cells were generated by replacement of the *DBP1* ORF with a cassette-encoding resistance to Hygromycin (*dbp1Δ::hphMX4*) (*Sen et al., 2019*). Consistent with our data, insertion of hphMX4 to replace the *DBP1* ORF caused mis-regulation of the adjacent gene *MRP51*. Despite levels of the Mrp51-encoding transcript remaining similar to wild-type in *dbp1Δ::hphMX4* cells, translation of *MRP51* was decreased 4.4-fold in these conditions (*Figure 1—figure supplement 1B,C*). The 3.7-fold decrease in TE of *MRP51* in *dbp1Δ::hphMX4* cells in this study led to the interpretation that the canonical *MRP51* transcript is highly dependent on Dbp1 for its translation (*Figure 1—figure supplement 1D*; *Sen et al., 2019*). A closer look at the transcripts produced from this locus, however, revealed the presence of a 5'-extended *MRP51* LUTI in the *dbp1Δ::hphMX4* cells, like the one we had observed. This 5' extended *MRP51* transcript contained competitive AUG-initiated uORFs that were translated in place of the *MRP51* ORF, as in our experiments (*Figure 1—figure supplement 1E*). These findings confirmed that the off-target effects seen in *dbp1Δ::kanMX6* mutants were not an artifact of our strain background, selection cassette, or experimental conditions.

## Selection-cassette-induced mis-regulation of *MRP51* leads to systemic phenotypic consequences

We next sought to determine whether the mutant phenotypes we observed in *dbp1Δ::kanMX6* cells were due to loss of *DBP1* function or resistance cassette-dependent mis-regulation of *MRP51*, a gene encoding a mitochondrial small subunit ribosome protein (mt-SSU) (*Green-Willms et al., 1998*). Polysome analysis of *dbp1Δ::kanMX6* cells during meiosis demonstrated that overall translation rates were diminished relative to wild-type cells, as determined by lower polysome peaks in the mutant (*Figure 2A*; right, fractions 5 and 6). Consistent with this finding, measurement of radioactive amino acid incorporation rates revealed a 24% decrease in bulk translation in *dbp1Δ::kanMX6* cells (*Figure 2—figure supplement 1A*). Polysome traces also indicated differences in the accumulation of a ribonucleoprotein (RNP) species roughly the size of the cytoplasmic large 60S subunit (*Figure 2A*; LSU, fraction 3) and a decrease in cytoplasmic small 40S subunit signal (*Figure 2A*; SSU, fraction 2). We performed label-free mass spectrometry on fractionated wild-type and *dbp1Δ::kanMX6* polysomes and used hierarchical clustering of the data to assess global differences in polysome composition. We found that a prominent cluster of proteins enriched in fraction 3 of polysomes from *dbp1Δ::kanMX6* cells compared to wild-type was highly enriched for the mitochondrial large ribosome subunit (mt-LSU). This cluster contained 34 of the 42 proteinaceous mt-LSU components quantified in our study (*Figure 2A*; left). Further analysis of fraction 3 revealed that every mitochondrial 54S protein quantified was enriched in *dbp1Δ::kanMX6* cells compared to the wild-type control (*Figure 2—figure supplement 1C*). These data suggest the accumulated RNP observed in *dbp1Δ::kanMX6* cells is the mt-LSU. A previous study from our laboratory observed that when individual cytoplasmic SSU proteins are lost, free cytoplasmic 60S LSUs accumulate, presumably a result of their inability to find an SSU to complex with and form fully assembled 80S species (*Cheng et al., 2019*). The build-up of free mt-LSU in *dbp1Δ::kanMX6* cells, which express lower amounts of mitochondrial small subunit (mt-SSU) component Mrp51 is reminiscent of that effect. Furthermore, we identified a small cluster of cytoplasmically translated mitochondrial proteins that were decreased in *dbp1Δ::kanMX6* high polysome fractions compared to wild-type, hinting at effects of broader mitochondrial dysfunction (*Figure 2A*). These data suggested the cassette-mediated mis-regulation of *MRP51* in *dbp1Δ::kanMX6* cells may have been responsible for the mutant phenotypes we observed, rather than loss of Dbp1 protein.

Disruption of mitochondrial translation, and ultimately function, would be expected to lead to cellular fitness defects. Based on the apparent mitochondrial defects observed in the mass spectrometry data, we hypothesized that *dbp1Δ::kanMX6* cells would grow poorly in conditions in which elevated mitochondrial function is required. Consistently, *dbp1Δ::kanMX6* cells exhibited severe

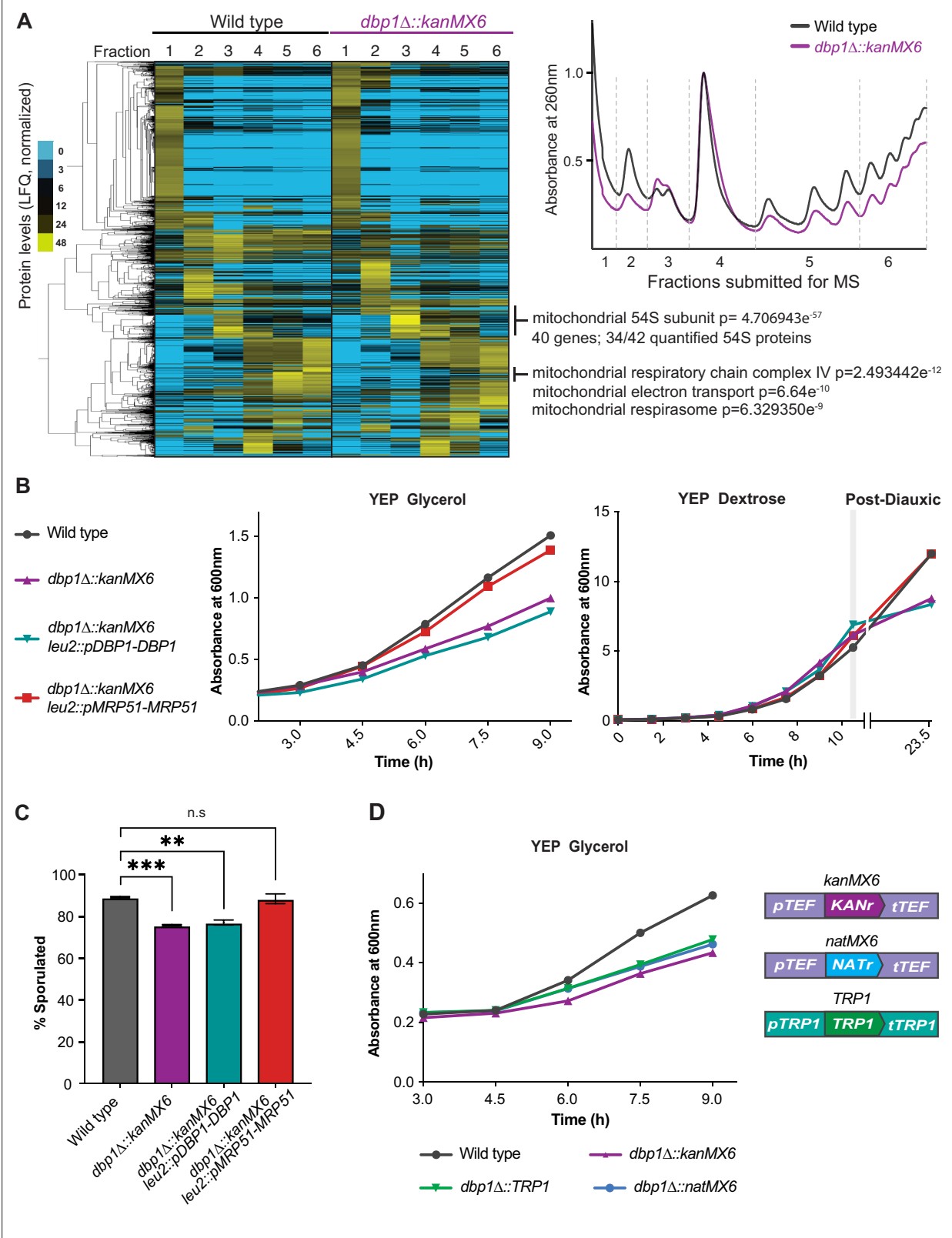

**Figure 2.** Cassette-driven mis-regulation of *MRP51* causes off-target mutant phenotypes. (**A**) Label-free mass spectrometry of fractionated polysomes from wild-type and *dbp1Δ::kanMX6* cells during meiosis (4 hr in sporulation media). Note decreased polysomes, accumulation of mitochondrial 54S subunit (fraction 3) in *dbp1Δ::kanMX6* cells, at right. Two biological replicates were analyzed, data from one are shown. Data were subjected to hierarchical clustering and normalized per protein. Mass spectrometry replicates and their agreement by spearman correlation are shown in *Figure 2—*

*Figure 2 continued on next page*

*Figure 2 continued*

**figure supplement 1B**. For entire dataset see **Figure 2—source data 1**. (**B, C**) Analysis of wild-type, *dbp1Δ::kanMX6*, *dbp1Δ::kanMX6 leu2::pDBP1-DBP1*, and *dbp1Δ::kanMX6 leu2::pMRP51-MRP51* cells under conditions where *dbp1Δ::kanMX6* mutant phenotypes were observed. (**B**) Representative growth curves in the non-fermentable carbon source glycerol (left), and ethanol (right: post-diauxic). Three or four independent biological replicates were performed for each experiment and data for all replicates are shown as doubling time (glycerol) or final culture absorbance (post-diauxic) in **Figure 2—figure supplement 2A,B**. (**C**) Twenty-four-hour sporulation efficiency counts, data shown are the average of three biological replicates with error bars showing SD. Statistical significance was determined by a one-way analysis of variance (ANOVA) adjusted for multiple comparisons with Dunnett's multiple comparison test where a reported **$p_{adj}$ < 0.005, and ***$p_{adj}$ < 0.0005. (**D**) Representative growth curves of wild-type and varied *dbp1Δ* cassette replaced mutants in the non-fermentable carbon source glycerol, cassette makeup is shown on the right. For this experiment four independent biological replicates were analyzed and the doubling time for each replicate is shown in **Figure 2—figure supplement 2E**.

The online version of this article includes the following source data and figure supplement(s) for figure 2:

**Source data 1.** Label-free quantification (LFQ) of all proteins quantified in fractionated polysome experiment shown in **Figure 2** and **Figure 2—figure supplement 1**.

**Figure supplement 1.** *DBP1* ORF replacement with kanMX6 causes off-target mutant phenotypes resulting from mitochondrial dysfunction.

**Figure supplement 2.** Cassette insertion at the *DBP1* locus causes broad off-target phenotypes that depend on Mrp51.

---

growth defects in media containing only non-fermentable carbon sources, such as glycerol or acetate (**Figure 2—figure supplement 1D**). Additionally, growth defects were evident in *dbp1Δ::kanMX6* cells grown in rich media (YEP dextrose or sucrose) during post-diauxic growth stages, after saturated cultures have exhausted fermentable carbon source availability and begin to utilize ethanol through respiration (**Figure 2—figure supplement 1E**). To determine whether these phenotypes were due to cassette-induced *MRP51* mis-regulation or loss of Dbp1 protein, we tested whether they could be rescued by exogenous expression of either Dbp1 or Mrp51. All *dbp1Δ::kanMX6* growth defects observed in conditions requiring elevated mitochondrial function remained similar with and without exogenous Dbp1 expression. In contrast, exogenous expression of Mrp51 rescued all previously observed respiratory growth defects of the *dbp1Δ::kanMX6* strain, confirming these phenotypes resulted from disrupted expression of Mrp51, rather than lack of Dbp1 protein (**Figure 2B**; **Figure 2—figure supplement 2A,B**).

We next assessed whether other observed phenotypes in *dbp1Δ::kanMX6* cells were due to off-target mis-regulation of *MRP51* or loss of Dbp1 protein. The sporulation defect observed in *dbp1Δ::kanMX6* cells was not rescued by exogenous Dbp1 but was fully rescued by exogenous Mrp51 expression (**Figure 2C**). Based on these results, we propose that this defect results from the dependency of meiotic cells on respiratory function (**Jambhekar and Amon, 2008**). Finally, polysome profiles of *dbp1Δ::kanMX6* cells exogenously expressing Mrp51 looked similar to wild-type polysomes while polysomes from *dbp1Δ::kanMX6* cells with or without exogenous expression of Dbp1 were indistinguishable (**Figure 2—figure supplement 2C**). In light of these data, we conclude that the reduced bulk cytoplasmic translation phenotypes observed in *dbp1Δ::kanMX6* cells reflected a cellular response to mitochondrial dysfunction due to off-target Mrp51 mis-regulation rather than loss of Dbp1 protein. Mitochondrial dysfunction and translation rates have both previously been shown to impact the TOR pathway and thus influence global translation (**Gao et al., 2016**; **Jazwinski, 2013**; **Raught et al., 2001**; **Topf et al., 2019**), consistent with our findings.

To ensure that the inability of Dbp1 to rescue phenotypes in *dbp1Δ::kanMX6* cells was not due to a lack of exogenous expression, we compared Dbp1 levels from the endogenous and exogenous loci. Western blot analysis revealed that the level of Dbp1 expressed from the exogenous integration was highly similar to that of the endogenous locus (**Figure 2—figure supplement 2D**). We conclude that resistance cassette insertion at the *DBP1* locus is sufficient to cause widespread off-target phenotypes by mis-regulation of Mrp51. These data raised concerns, given the widespread use of expression cassette insertion in genome editing and were consistent with the phenomena of the NGE (**Ben-Shitrit et al., 2012**; **Atias et al., 2016**; **Egorov et al., 2021**), identified by analysis of deletion collection mutant phenotypes. While these results were disappointing for our study of the function of Dbp1 helicase, they highlighted this locus as a useful context to dissect the mechanism behind, and ideally to fix, these poorly understood cassette-related side effects.

To assess whether cassette-driven off-target effects were common to a specific feature of the kanMX6 and hphMX4 cassettes, we replaced the *DBP1* ORF in wild-type cells with two additional cassettes, including one (natMX6; **Figure 2D**) that shared the *pTEF* and *tTEF* regulatory regions with

kanMX6 and hphMX4 and one (*TRP1*; *Figure 2D*) that did not and is relatively lowly expressed. Insertion of either additional cassette led to a growth defect in non-fermentable carbon source compared to wild-type cells that was similar to that seen with kanMX6 insertion (*Figure 2D*, *Figure 2—figure supplement 2E*). As all reported cases of NGE have represented gene replacements with the commonly used kanMX modules, it was previously unclear whether this effect simply represented a problematic feature of kanMX. Our data indicate that the aberrant-transcript-driven mis-regulation of *MRP51* is a general consequence of expression cassette insertion rather than a specific result of any sparticular cassette feature .

## Transcription-terminator-flanked cassettes prevent adjacent gene mis-regulation

We reasoned that if the expression cassette inserted at the *DBP1* locus drove transcription that altered neighboring gene expression, 'insulation' of neighboring genomic regions from transcriptional activity of the cassette should prevent these off-target effects. Toward this end, we placed strong transcription terminator sequences flanking both ends of the resistance cassette (*Song et al., 2016*). This included placing a portion of the *DEG1* terminator 5′ to the *TEF* promoter, and either the same *DEG1* terminator or the *CYC1* terminator sequence 3′ to the *TEF* terminator within the kanMX6 cassette (*Figure 3A*). We named these plasmids 'kanMX6-ins1' and 'kanMX6-ins2' and used them to replace *DBP1* using the same primers and insertion location as before. Both *dbp1Δ::kanMX6-ins* strains grew at wild-type rates in the non-fermentable carbon source, glycerol, in contrast to cells housing the original cassette replacement (*Figure 3B*, *Figure 3—figure supplement 1A*). Consistent with this finding, levels of Mrp51 protein in the *dbp1Δ::kanMX6-ins* strains were similar to those in wild-type cells (*Figure 3C, D*). Finally, 5′ rapid amplification of cDNA ends (5′RACE) confirmed that the *MRP51* transcript produced in the *dbp1Δ::kanMX6-ins* strains was the same as that seen in wild-type cells, as opposed to the synthetic *MRP51* LUTI in the initial cassette-inserted strains (*Figure 3E*, *Figure 3—figure supplement 1B,C*). To facilitate the use of this strategy and prevent aberrant transcription and neighboring gene mis-regulation in future studies, we created three additional 'insulated' cassettes housing the selection markers *TRP1*, *his5⁺* (*S. pombe HIS5*; which complements *S. cerevisiae his3*), and *natʳ* (nourseothricin resistance) within the classic 'MX6' backbone (*Longtine et al., 1998*; *Wach et al., 1997*; *Wach et al., 1994*; *Figure 3F*).

While flanking both ends of the cassette with transcription terminator sequences was sufficient to prevent off-target mutant phenotypes at the *DBP1/MRP51* locus, we wished to identify the root cause of neighboring gene mis-regulation. Previous work had suggested that the unusually high expression level of the kanMX6 cassette could cause gross chromatin changes that lead to local changes in transcription, and the NGE (*Ben-Shitrit et al., 2012*; *Egorov et al., 2021*). However, the similar off-target respiratory growth phenotype observed when *DBP1* was replaced with either the *kanMX6* or *TRP1* cassettes, driven by the *A. gossypii TEF* and *S. cerevisiae TRP1* promoters, respectively, led us to disfavor this model, as *pTRP1* expression is several-fold lower than *pTEF*. Because bidirectional transcription has been shown to occur from most, if not all, promoters (*Neil et al., 2009*; *Xu et al., 2009*; *Wei et al., 2011*; *Teodorovic et al., 2007*; *Preker et al., 2008*; *Core et al., 2008*; *Seila et al., 2008*), and the observed aberrant transcript originated from the promoter-proximal end of the cassette, we hypothesized that bidirectional promoter activity from the *TEF* promoter was the most likely cause of *MRP51* LUTI-based repression. Consistent with this model, both *S. cerevisiae pTEF1* and *pTRP1* display bidirectional activity at their endogenous loci, as determined by tiling array analysis and nascent transcript sequencing (NETseq) (*Xu et al., 2009*; *Churchman and Weissman, 2011*). We first tested this model by inserting a kanMX6 cassette in the reverse orientation at the *DBP1* locus. This would not be expected to prevent gross chromatin changes caused by extremely high expression of the *kanʳ* gene but would move *pTEF* away from *MRP51* (*Figure 3G*, *Figure 3—figure supplement 2A*). Indeed, we found that replacing *DBP1* with the reversed *kanMX6* cassette did not result in a respiratory growth defect (*Figure 3G*; *Figure 3—figure supplement 2A*). Considering these data, and the knowledge that shared homology between cassettes can reduce the efficiency of targeted insertions when used sequentially, we constructed two more 'insulated' cassettes containing only a single additional terminator (*tDEG1* or *tCYC1*) (*Song et al., 2016*) flanking each promoter and sharing no regulatory sequences with each other. Furthermore, to prevent the possibility of recombination between endogenous sequences and these new cassettes, we used the *PGK1* promoter and

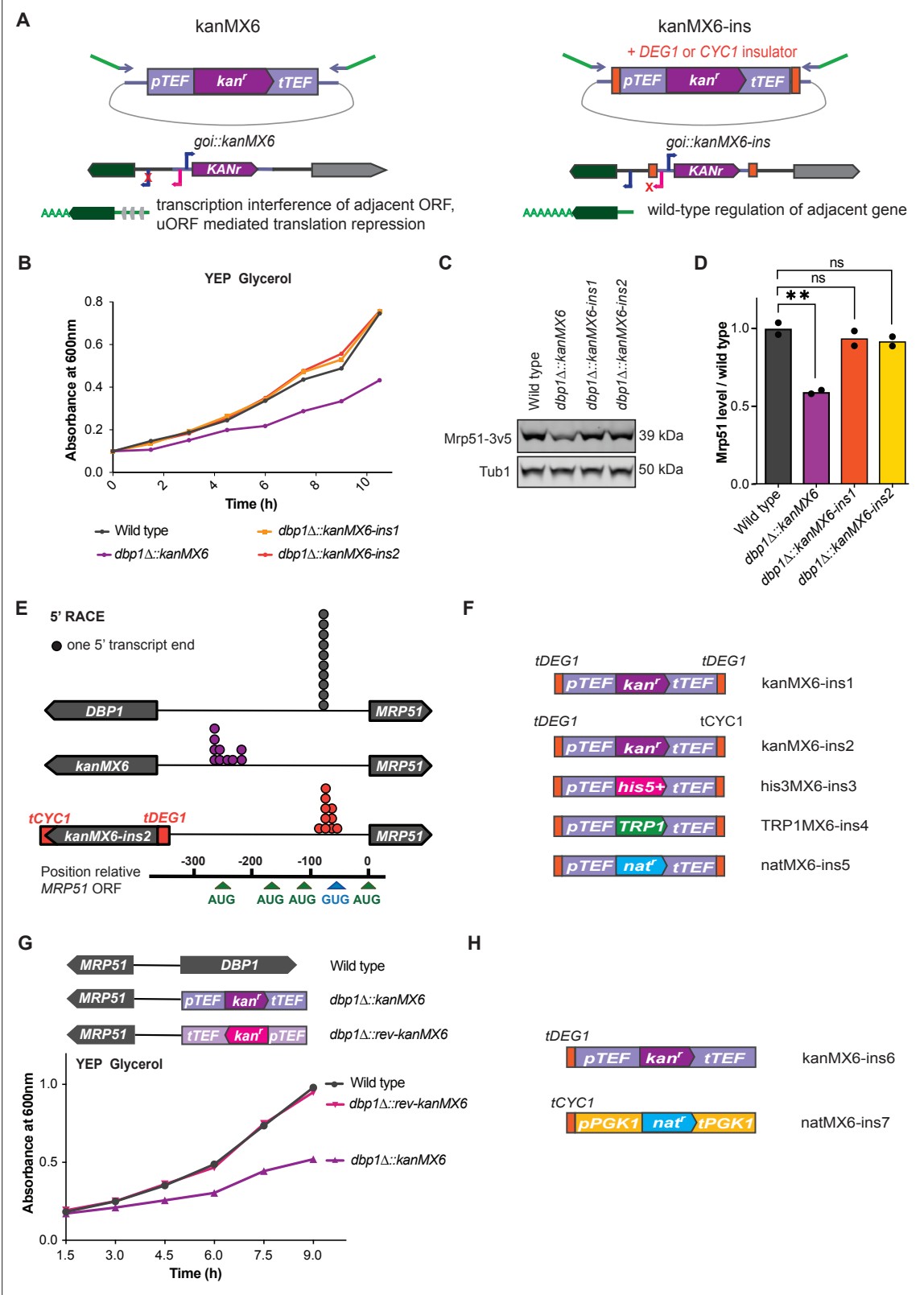

**Figure 3.** Transcription-terminator-flanked cassettes prevent adjacent gene mis-regulation. (**A**) Design of 'insulated' kanMX6-ins cassettes and proposed model. The *DEG1* transcription terminator sequence was inserted 5' of the *TEF* promoter and either the *DEG1* or *CYC1* terminator was inserted 3' of the *TEF* terminator in pFA6a-kanMX6 (***Longtine et al., 1998***). "goi" in the cartoon is an abreviation for "gene of interest". (**B**) A single representative trace showing growth of *dbp1Δ::kanMX6* and *dbp1Δ::kanMX6-ins* strains compared to wild-type in YEP glycerol. Three independent replicates

*Figure 3 continued on next page*

*Figure 3 continued*

were performed and the doubling time for each replicate is shown in *Figure 3—figure supplement 1A*. (**C**) Western blots for Mrp51-3v5 levels in *dbp1Δ::kanMX6* and *dbp1Δ::kanMX6-ins* cells compared to wild-type during mitotic exponential growth in rich media. For full blot scans see *Figure 3— source data 1*. (**D**) Quantification of western blots as in (**C**), with Mrp51 normalized to Tub1 (alpha tubulin). Data shown are two independent biological replicates. Statistical significance was determined by a one-way analysis of variance (ANOVA) adjusted for multiple comparisons using Dunnett's multiple comparisons test where **$p_{adj}$ < 0.005. (**E**) 5′ rapid amplification of cDNA ends (RACE) demonstrates the 5′ ends of transcripts produced in wild-type, *dbp1Δ::kanMX6*, and *dbp1Δ::kanMX6-ins* strains. Gel demonstrating 5′ end products sequenced for each sample and the exact 5′ mRNA end sequences listed in *Figure 3—figure supplement 1B,C*. (**F**) Schematics of improved selection cassettes cloned with terminators insulating the 5′ and 3′ cassette ends. (**G**) Top: schematic of the reversed orientation cassette inserted at the *DBP1* locus. Bottom: a single representative growth curve of wild-type, *dbp1Δ::kanMX6*, and *dbp1Δ::rev-kanMX6* cells grown in the non-fermentable carbon source glycerol. Four independent biological replicates were analyzed and doubling times for each replicate are shown in *Figure 3—figure supplement 2A*. (**H**) Design of two additional cassettes with minimal sequence overlap, to enable their paired use in the same strain, and insulated only on the 5′ cassette end. Growth data for these new cassettes in *Figure 3—figure supplement 2B*.

The online version of this article includes the following source data and figure supplement(s) for figure 3:

**Source data 1.** Zipped folder containing uncropped tif image of western blot shown in *Figure 3* with and without labels on the relevant samples and bands.

**Figure supplement 1.** *DBP1* ORF replacement with kanMX6-ins cassettes yields wild-type *MRP51* transcripts and growth rates.

**Figure supplement 1—source data 1.** Zipped folder containing uncropped tif image of agarose gel showing the amplified 5′ RACE products sequenced in *Figure 3* and shown in *Figure 3—figure supplement 1*.

**Figure supplement 2.** *MRP51* mis-regulation is caused by cassette promoter-driven divergent transcription.

terminator sequence from *C. glabrata* to drive resistance ORF expression (*Figure 3H*). Use of these two new insulated cassettes to replace *DBP1* in the forward and reverse orientation confirmed that neither cassette caused a respiratory growth defect, in either orientation (*Figure 3—figure supplement 2B*).

## Cassette insertion drives the production of divergent transcripts at all loci but local sequence features determine their length and stability

Though severe off-target effects driven by a stable divergent transcript were observed with cassette replacement of *DBP1*, cassette replacement occurs at many loci without neighboring gene disruption. To better understand how the mis-regulation occurring with cassette insertion at the *DBP1* locus compared to effects at other loci, we analyzed mRNA-seq and ribosome profiling data over genomic intervals surrounding cassette insertions from 18 additional cassette-replacement mutant datasets from our laboratory (*Cheng et al., 2019*), and one case in which only mRNA-seq data were available. All analyzed mutants contained a single ORF replaced with either the kanMX6 or natMX6 resistance cassettes and samples were collected during mitotic exponential growth. mRNA-seq data revealed stable divergent transcripts stemming from cassette-mediated gene replacements at 5 of these 19 loci (*Figure 4A–E*). For all five of these examples, the aberrant transcript was driven from the 5′ end of the resistance cassette, like in *dbp1Δ::kanMX6* cells (Figure 1E; *Figure 4A–E*), and consistent with a model whereby bidirectional transcription from the cassette promoter drives aberrant neighboring transcripts. However, unlike the strong mis-regulation of *MRP51* seen with *DBP1* ORF replacement, these divergent transcription events had no obvious effect on the translation of adjacent genes, and a lack of apparent uORF translation in the extended 5′ mRNA regions (*Figure 4A–D*). For four of these new cases in which aberrant transcription resulted from a cassette-based deletion, the adjacent gene was positioned in the same orientation as the aberrant TSS thus it would be possible for synthetic LUTI-based repression to occur as was observed for *MRP51*. Although, we also note that for three of these four cases, expression of the adjacent ORF was low, even in wild-type cells, and thus mis-regulation of these genes may be difficult to detect under these conditions (*Figure 4B–D*). For the last case, in which *UPF1* was replaced with the natMX6 cassette, an abundant cassette-induced divergent transcript was observed. However, the adjacent *ISF1* gene was oriented antisense to this transcript and there was no evidence of its transcription with or without natMX6 insertion at *UPF1* (*Figure 4E*); hence, there was no baseline expression to disrupt. Together, these data suggest that stable cassette-induced divergent transcripts are fairly common (occurring at 6/20 of loci analyzed here), but that severe phenotypic consequences are more rare.

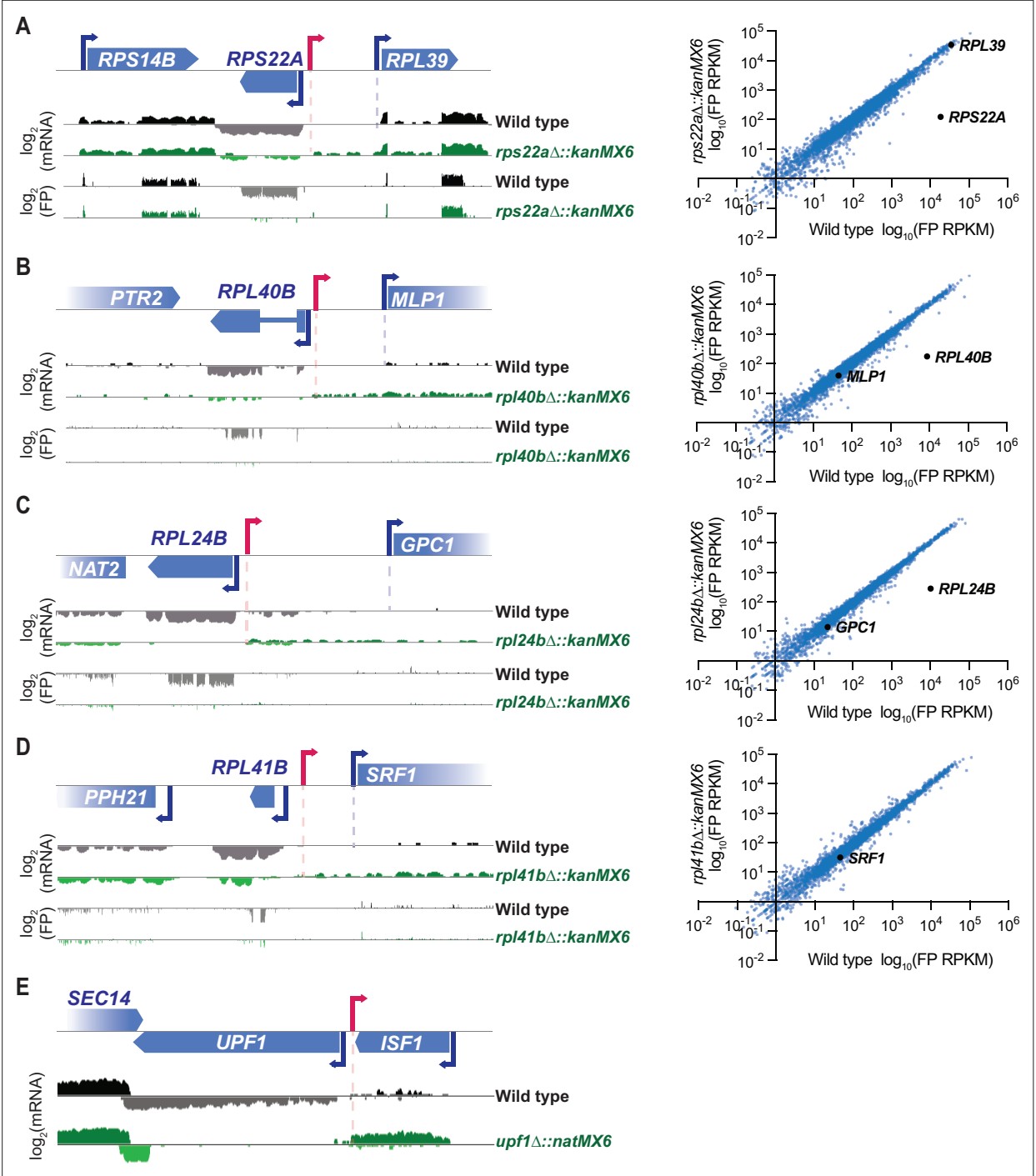

**Figure 4.** Stable resistance cassette-induced divergent transcripts are observed at ~30% of inserted loci. (**A–D**) 18 ribosomal protein deletions generated by ORF replacement with kanMX6 from *Cheng et al., 2019* were observed for cassette-driven divergent transcription events. Left, ribosome profiling footprint (FP) and mRNA-seq reads aligned to the ORF-replaced locus for each strain. Right, FP RPKM for every ORF expressed in the mutant and wild-type strains with the deleted ORF and its potentially disrupted genomic neighbor marked in black. Data were collected from vegetative exponentially growing cells. (**D**) Note: *RPL41B* is not quantified in translation graph because the ORF is only 25 amino acids long. (**E**) mRNA-seq reads aligned to the replaced locus in a *upf1Δ::natMX6* mutant. Data were collected from vegetative exponentially growing cells. For raw data see *Figure 4— source data 1*. (**A–E**) Data shown here represent a single biological replicate for each experiment and pink arrows denote likely divergent transcription start site (TSS).

The online version of this article includes the following source data for figure 4:

**Source data 1.** Zipped folder containing wig files for mRNA-seq reads surrounding the *UPF1* locus in wild-type and *upf1Δ::natMX6* cells.

We noted that the divergent transcript observed with cassette insertion at the *UPF1* locus (*Figure 4E*), which encodes a gene that drives nonsense-mediated RNA decay (NMD) (*Hug et al., 2016*), was more abundant than any other example we found. This combined with the lack of stable divergent transcripts at ~70% of cassette-inserted loci, led us to wonder if RNA degradation could mask aberrant transcription events. Although NMD might be expected to target a subset of aberrant transcripts that reach the cytosol, nuclear exosome activity seemed likely to be a more major contributor (*Hug et al., 2016*; *Gudipati et al., 2012*; *Wyers et al., 2005*). In fact, one of the studies that initially identified bidirectional transcription as a feature of eukaryotic promoters relied on cells lacking nuclear exosome activity by loss of function of the nuclear catalytic subunit Rrp6 (*Xu et al., 2009*). It subsequently became clear that an alternative transcription termination mechanism, the 'NNS' pathway—named for complex subunits Nrd1, Nab3, and Sen1—leads to termination of most divergent transcripts and their rapid degradation by the Rrp6/exosome (*Gudipati et al., 2012*; *Porrua and Libri, 2015*; *Wyers et al., 2005*; *Schulz et al., 2013*; *Steinmetz et al., 2001*).

To test if Rrp6/exosome-mediated degradation was masking divergent transcripts produced from cassette insertion, we analyzed published global mRNA data from strain backgrounds with cassette insertions in an *rrp6Δ* strain background. For four cassette-replaced loci, control data were also available for the cassette replacement in the presence of *RRP6* (*Figure 5A–D*; *Malabat et al., 2015*; *Wang et al., 2020*; *Rege et al., 2015*). For all of these cases, a divergent transcript from the site of cassette insertion was observed in *rrp6Δ* cells that either was absent or less abundant in the *RRP6* control (*Figure 5A–D*). Interestingly, this included a case in which *UPF1* was deleted by cassette insertion and 5′ transcript ends were sequenced (*Figure 5D*; *Malabat et al., 2015*). Here, the divergent transcript's 5′ end was observed in the *RRP6* control, consistent with our data, but was even more abundant in *rrp6Δ* cells (*Figure 4E*; *Figure 5D*). In the three other cases, for which data were available for cassette insertions in a *rrp6Δ* background but not a *RRP6* control strain, including data for a *rrp6Δ::kanMX6* mutant itself, cassette-induced divergent transcripts were apparent (*Figure 5—figure supplement 1A–C*; *Wang et al., 2020*; *Schmidt et al., 2012*). Thus, for 7/7 (100%) of loci, replaced with a variety of cassettes, divergent transcripts were observed from inserted cassettes when nuclear exosome function was absent (achieved by *rrp6Δ*; *Figure 5A–D*, *Figure 5—figure supplement 1A–C*). Although this sample size is not large (*n* = 7: *rrp6Δ*, *n* = 20: *RRP6*), this result supports the model that expression cassette insertion always leads to production of a divergent transcript, driven by bidirectional promoter activity, but that these transcripts are usually terminated and degraded in cells with normal nuclear exosome activity. Together, these data argue that all loci are susceptible to cassette-induced bidirectional-promoter-driven production of divergent transcripts, but additional local characteristics at many loci prevent divergent transcription from disrupting neighboring genes.

Placement of large regions of DNA from other yeast species into *S. cerevisiae* results in an increased amount of divergent transcription from the foreign promoters, relative to endogenous *S. cerevisiae* promoters in the same cells (*Jin et al., 2017*). Because the *pTEF* version in commonly used budding yeast expression cassettes, including kanMX, is derived from *A. gossypii*, we wondered if species-specific regulation exacerbates the divergent transcription that we observe with cassette insertion (*Wach et al., 1994*). To test this, we swapped the *pTEF* in kanMX6, with the homologus *S. cerevisiae pTEF1* region and inserted the new cassette at the *DBP1* locus (*Figure 5E*; short-pTEF1). Insertion of the *S. cerevisiae TEF1* promoter caused an even stronger respiratory growth defect than was observed with the original *A. gossypii pTEF* (*Figure 5E*, *Figure 5—figure supplement 1D*). This result is consistent with our finding that the *S. cerevisiae TRP1* promoter can drive divergent transcription and neighboring gene disruption (*Figure 2D*, *Figure 2—figure supplement 2E*), and argues that off-target effects stemming from expression cassette insertion are not solely a result of inserting a foreign promoter sequence into *S. cerevisiae*.

Even at its endogenous locus, *S. cerevisiae pTEF1* drives a divergent transcript (cryptic unstable transcript 910; '*CUT910*') that is unstable due to Rrp6/exosome activity (*Figure 5E*; *Xu et al., 2009*), which degrades NNS-terminated divergent transcripts (*Fox et al., 2015*). The early termination and degradation of *CUT910* presumably protects the neighboring locus *MRL1* from transcription interference (*Xu et al., 2009*). We thus hypothesized that neighboring gene disruption by divergent transcripts resulting from bidirectional *pTEF1* (or *pTEF*) activity at expression cassette insertion sites is partially a result of the use of a 'minimal' promoter region. While this region is competent to drive transcription of a transgene, it is devoid of its larger natural context and evolved failsafe mechanisms,

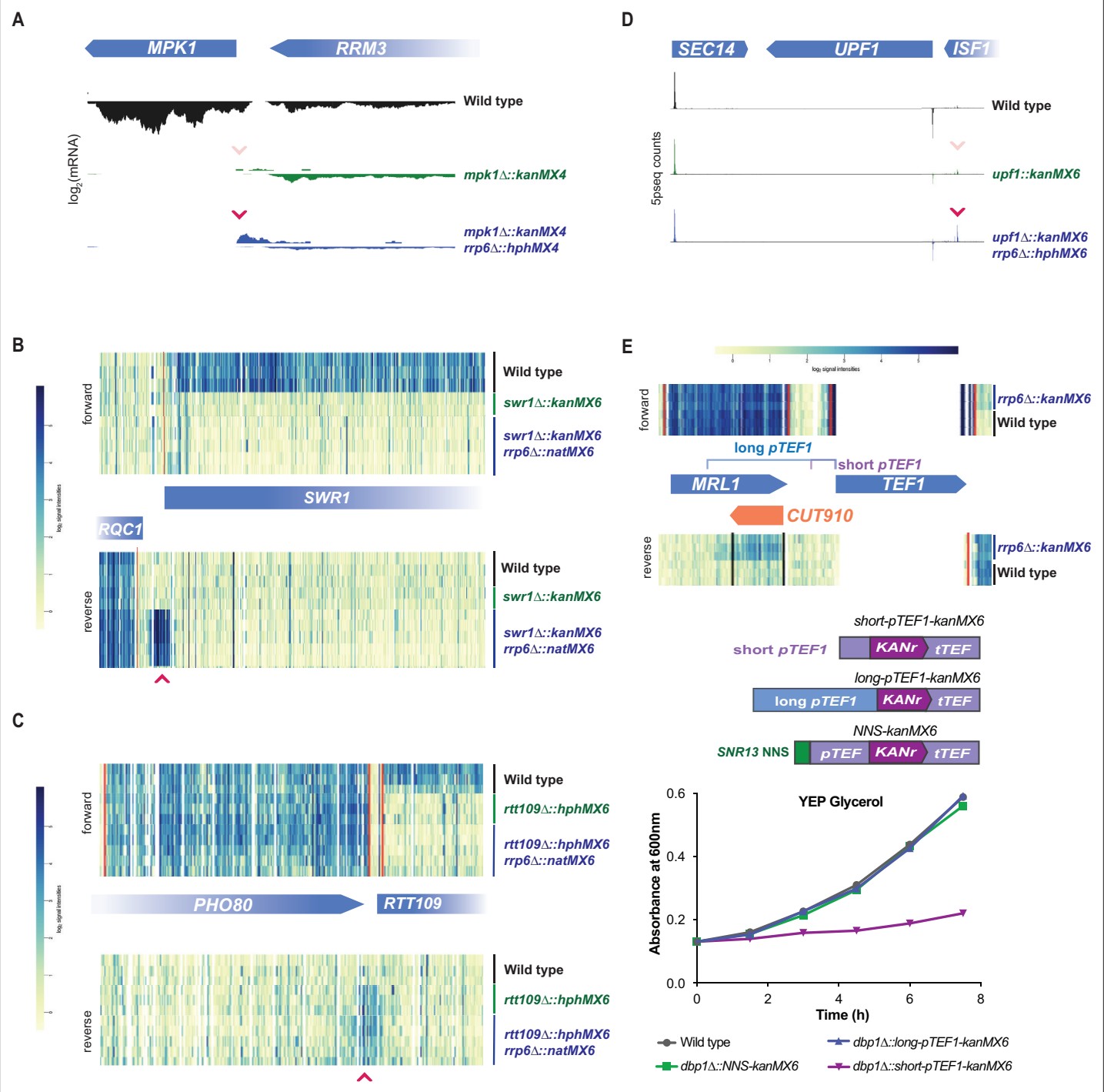

**Figure 5.** Rrp6-mediated decay masks divergent transcripts produced from expression cassette promoters at most loci. (**A–D**) Global mRNA data for cassette-inserted mutants in the presence and absence of Rrp6-mediated RNA decay. Pink arrows indicate cassette-driven divergent transcripts enriched in *rrp6Δ* backgrounds. (**A**) mRNA-seq data for wild-type, *mpk1Δ::kanMX4*, and *mpk1Δ::kanMX4 rrp6Δ::hphMX4* cells from **Wang et al., 2020** are shown over the genomic region including *MPK1* and its neighboring gene. (**B**) Tiling array data for wild-type, *swr1Δ::kanMX6*, and *swr1Δ::kanMX6 rrp6Δ::natMX6* cells from **Rege et al., 2015** are shown over the genomic region including *SWR1* and its neighboring gene. (**C**) Tiling array data for wild-type, *rtt109Δ::hphMX6*, and *rtt109Δ::hphMX6 rrp6Δ::natMX6* cells from **Rege et al., 2015** are shown over the genomic region including *RTT109* and its neighboring gene. (**D**) 5prime-seq data for wild-type, *upf1Δ::kanMX6*, and *upf1Δ::kanMX6 rrp6Δ::hphMX6* cells from **Malabat et al., 2015** are shown over the genomic region including *UPF1* and its neighboring genes. (**E**) Top: tiling array data for wild-type and *rrp6Δ::kanMX6* cells from **Xu et al., 2009** are shown for the genomic region surrounding *S. cerevisiae pTEF1*. The *TEF1* ORF shares high sequence homology with *TEF2*, leading to the lack of unique mapped reads in that region. Position of the natural divergent unstable transcript (*CUT910*) in relation to the long and short *pTEF1* regions

*Figure 5 continued on next page*

*Figure 5 continued*

used below. Middle: schematics of constructs including the short and long *S. cerevisiae pTEF1* regions in place of *pTEF* in the *kanMX6* cassette are shown, as is a version of *kanMX6* in which the NNS termination site for *SNR13* is inserted flanking *pTEF*. Bottom: representative growth curves of wild-type, *dbp1Δ::long-pTEF1-kanMX6*, *dbp1Δ::short-pTEF1-kanMX6*, and *dbp1Δ::NNS-pTEF-kanMX6* cells are shown in YEP glycerol. Four independent replicates were analyzed and calculated doubling time for each replicate is shown in *Figure 5—figure supplement 1D*.

The online version of this article includes the following source data and figure supplement(s) for figure 5:

**Figure supplement 1.** Cells lacking Rrp6 express stable divergent transcripts from all inserted cassette promoters.

**Figure supplement 2.** Cas9-mediated cassette-free deletion of *DBP1* ORF causes aberrant translation of an ORF within the *dbp1Δ* 3'UTR.

**Figure supplement 2—source data 1.** RPKM values for all genes quantified in ribosome profiling experiment shown in *Figure 5—figure supplement 2*.

including NNS-based termination signals, which prevent bidirectional transcripts from interfering with neighboring loci (*Schulz et al., 2013*; *Porrua and Libri, 2015*). Indeed, we found that when the minimal *short-pTEF1* was replaced with a longer region of the endogenous promoter region, including the predicted termination site for *CUT910* (*Xu et al., 2009*; *Figure 5E*; *long-pTEF1*), the respiratory growth defect seen with *short-pTEF1* insertion was alleviated (*Figure 5E*). We concluded that the off-target effect from cassette insertion is a result of porting a minimal *pTEF1* out of its native context, which leads to unmasking, in some insertion sites, of strong divergent transcript production.

These data suggested that naturally occurring NNS termination sites may be an important contributing factor in determining which loci are sensitive to the cassette-driven NGE. For many loci, the cassette-driven divergent transcripts observed in the *rrp6Δ* strains were short, likely reflecting termination by the NNS pathway. Though short binding preference motifs have been determined for the NNS complex, predictions of naturally occurring NNS termination signals based on sequence alone is difficult (*Porrua and Libri, 2015*; *Carroll et al., 2004*). Therefore, to test the sufficiency of the NNS termination pathway to terminate and prevent cassette-induced neighboring gene disruption, we tested whether a validated NNS termination site flanking *pTEF* in the kanMX6 cassette could reduce the impact of divergent transcription caused by cassette insertion at *DBP1*. We found that the ~150 bp intergenic region directly following mature *SNR13* RNA, which has been shown to induce NNS-based termination in exogenous transcripts, was sufficient to protect the *MRP51* locus from cassette-induced LUTI-based repression (*Steinmetz et al., 2001*; *Schaughency et al., 2014*). This was demonstrated by the wild-type growth rates observed in non-fermentable carbon sources when the NNS-kanMX6 cassette was inserted at the *DBP1* locus (*Figure 5E*).

## Discussion

In this study, we provide evidence that genome-inserted expression cassettes drive bidirectional transcription and the production of divergent transcripts that can repress neighboring genes in yeast, resulting in potent off-target effects, even in carefully designed single-gene studies. Our data also provide a mechanism for the mysterious NGE, identified in analyses of yeast deletion collection studies, and clarify why some loci are susceptible to off-target effects, while some are immune (*Ben-Shitrit et al., 2012*; *Atias et al., 2016*; *Egorov et al., 2021*; *Makeeva et al., 2019*). We find that expression cassette insertion at *DBP1* reduces the translation of neighboring gene *MRP51* through a divergent LUTI driven by the bidirectional activity of cassette promoters (*Neil et al., 2009*; *Xu et al., 2009*; *Churchman and Weissman, 2011*). Production of this divergent transcript drives both transcription interference and uORF-based translation repression for *MRP51* (*Chia et al., 2017*; *Chen et al., 2017*; *Cheng et al., 2018*). Because most, if not all, eukaryotic promoters are thought to be bidirectional, these data suggest that any inserted expression cassette containing a promoter could drive neighboring off-target effects (*Teodorovic et al., 2007*; *Neil et al., 2009*; *Preker et al., 2008*; *Core et al., 2008*; *Xu et al., 2009*; *Seila et al., 2008*).

How commonly does expression cassette insertion cause confounding mutant phenotypes? Confidence in the specificity of yeast genome-editing cassettes has motivated their widespread use, and many landmark studies defining the functions of conserved proteins have been performed using this approach. It is routine for multiple loci to be disrupted within the same strain to enable analysis of genetic interactions, and global yeast deletion, hypomorphic, and epitope tag collections were created by use of the kanMX insertion cassette (*Giaever et al., 2002*; *Winzeler et al., 1999*; *Giaever and*

*Nislow, 2014*; *Chu and Davis, 2008*; *Ghaemmaghami et al., 2003*; *Huh et al., 2003*; *Breslow et al., 2008*). Despite the first reports of NGEs over 10 years ago, the original yeast cassettes remain widely used (*Ben-Shitrit et al., 2012*). One study that analyzed mutants generated with kanMX modules estimated that replacement at as many as 20% of loci may cause neighboring gene mis-regulation, and ~10% of deletion collection strains have been estimated to demonstrate non-specific phenotypes in genetic interaction data that could be attributed to neighboring gene mis-regulation (*Ben-Shitrit et al., 2012*; *Atias et al., 2016*; *Egorov et al., 2021*). A recent study focused on protein quantification of each yeast deletion collection strain found that one of the most significant factors determining whether a specific protein's abundance would change following deletion of a gene encoding another protein was whether the genes encoding the protein pair were genomic neighbors (*Messner et al., 2022*). It is important to note that all previously reported examples of the NGE, including those noted here, focused only on integrations of kanMX modules (*Ben-Shitrit et al., 2012*; *Atias et al., 2016*; *Egorov et al., 2021*; *Makeeva et al., 2019*; *Messner et al., 2022*). Our study determined that NGEs are driven by cassettes of disparate sequence. This observation allowed us to determine that the promoter bidirectionality that is inherent to eukaryotic promoters, rather than any feature of a particular cassette sequence, drives neighboring gene disruption. Thus, any expression cassette can cause disruptive NGEs, suggesting a much larger pool of potentially affected mutants than previously considered.

In fact, we found cassette-driven divergent transcripts to be produced at 100% of expression cassette-inserted loci analyzed, but they were often short and masked by exosome-mediated RNA decay. This suggests that additional features local to cassette insertion can prevent off-target phenotypes. Given that bidirectional promoter activity originating from cassettes drives neighboring gene disruption, there are two non-mutually exclusive explanations for why some loci would produce disruptive divergent transcripts and others would not. First, the different chromatin context at inserted loci may affect the bidirectional promoter activity (*Churchman and Weissman, 2011*; *Jin et al., 2017*; *Struhl, 1985*; *Porrua and Libri, 2015*; *Marquardt et al., 2014*). Alternatively, bidirectional activity and divergent transcript production could occur at all loci, with off-target effects being prevented in many cases by local sequence characteristics. Our data do not exclude the possibility that chromatin context is a factor, but do suggest that the role of local sequence features in determining the span of divergent transcription is important. One such feature that may naturally prevent neighboring gene disruption at cassette-inserted loci are NNS termination signals, like those known to terminate and prevent endogenously produced divergent transcripts from interfering with neighboring genes (*Steinmetz et al., 2001*; *Schulz et al., 2013*; *Porrua and Libri, 2015*). Both NNS termination signals and canonical coding gene transcription termination sequences were sufficient to prevent to 'insulate' bidirectional promoters within selection cassettes inserted at the *DBP1* locus.

How do divergent transcripts from cassette-inserted loci drive off-target phenotypes? While we focused on the integrated transcription and translation regulation driven by LUTI-based disruption of *MRP51* in this study, mis-regulation of adjacent genes by cassette-driven transcription interference alone may be even more frequent. LUTI-based neighboring gene disruption requires the adjacent gene to be in a 'head-to-head' orientation, the presence of repressive uORFs upstream of the coding sequence, and a continuous divergent transcript containing the entire ORF of the repressed gene (*Chia et al., 2017*; *Chen et al., 2017*; *Cheng et al., 2018*). Cassette-driven transcription interference, however, can occur regardless of the presence of repressive uORFs upstream of the neighboring coding sequence, does not require the divergent transcript to be continuous over the neighboring ORF, and can act on genes in either orientation to the inserted cassette (*Chia et al., 2017*; *Hirschman et al., 1988*; *Martens et al., 2004*; *van Werven et al., 2012*; *Hongay et al., 2006*; *Hausler and Somerville, 1979*; *Adhya and Gottesman, 1982*; *Proudfoot, 1986*; *Boussadia et al., 1997*; *Emerman and Temin, 1984*; *Corbin and Maniatis, 1989*). For example, in haploid cells, induction into meiosis is blocked by the expression of two separate non-coding RNAs that repress an antisense mRNA (*RME2/IME4*) or a downstream sense RNA (*IRT1/IME1*) via transcription interference (*Hongay et al., 2006*; *van Werven et al., 2012*). Transcription interference between loci can be effective over large distances, like the ~1800 nt span over which transcription of the repressive *IRT1* long non-coding RNA represses *IME1* (*van Werven et al., 2012*).

The type of cassette-mediated off-target effect in this study is difficult to detect or predict by sequence alone. No unexpected mutations exist in these cases and even mRNA measurements of

neighboring genes are not diagnostic unless all possible transcript isoforms are measured from a given locus. Additionally, our data suggest that at as many as 70% of cassette-inserted loci, divergent transcripts are produced but degraded by the nuclear exosome, meaning that even the lack of an observable stable divergent transcript is not sufficient to conclude wild-type regulation from a cassette-adjacent gene because disruption can occur regardless of the transcript's stability. Measuring translation or protein levels for adjacent genes should be diagnostic of this undesirable side effect, but ribosome profiling is not a routine experiment that is applied to most mutants, and tagging proteins can alter function and regulation. Additionally, as many genes are expressed in a condition-specific manner, measurement of the adjacent gene's protein levels compared to wild-type may give false confidence unless the proper conditions are chosen. Consistently, we observed condition-specific manifestation of the cassette-driven divergent transcription at the *DBP1/MRP51* locus. For 3/5 additional potential NGE cases with detectable aberrant transcripts in exosome proficient cells (*Figure 4*), adjacent genes were lowly expressed, even in wild-type cells under the conditions surveyed. Considering the number of factors affecting whether cassette adjacent genes are mis-regulated, we argue that it is currently not possible to predict when and where cassette insertion would result in mis-regulation severe enough to mislead studies.

Despite this complexity, complementation-based rescue experiments—namely single-copy insertion of the deleted gene with its entire endogenous regulatory region at another locus—allowed us to identify the unexpected regulation in the case of *DBP1* disruption, and should be more commonly used to confirm phenotypes. However, for genome-wide experiments, those using cassettes for epitope tagging, or experiments in which genetic interactions between several genes are being investigated, this may not be feasible. Therefore, we created 'insulated' resistance cassettes containing transcription terminator sequences flanking the promoter, which prevent off-target effects and possible misinterpretation of mutant phenotypes. Other possible alternatives to avoiding these confounding off-target effects include methods that allow mutants to be isolated without expression cassette insertion. Seamless deletions such as those created with the CRISPR/Cas9 approach avoid cassette insertions, but can still result in changes to the local environment of adjacent genes, potentially resulting in off-target effects (*Jinek et al., 2012*; *Güldener et al., 1996*; *Gray et al., 2005*). For the case of a markerless CRISPR/Cas9-based deletion of the *DBP1* ORF, we observed translation of a previously untranslated ORF within the mutant transcript containing only the *DBP1* 5′ and 3′ UTR (*Figure 5—figure supplement 2*). One could imagine that in cases like these, translation of novel ORFs could also cause misleading neomorphic mutant phenotypes, not linked to loss of function of the deleted ORF. These data highlight the point that even well controlled genomic changes can have unintended consequences on the translation of neighboring coding sequences, and emphasize the value of careful analysis of gene expression from loci neighboring those that are manipulated.

We note that two-step strategies to isolate mutants free of selection cassettes have been described but are often used the context of allowing subsequent re-use of auxotrophic markers and cassettes (*Güldener et al., 1996*; *Gray et al., 2005*). One example of this uses the Cre/loxP system to excise resistance cassettes following mutation isolation (*Güldener et al., 1996*). While this strategy can prevent neighboring gene disruption by cassettes, it is slower and more labor intensive than a single step process, and thus may not be commonly utilized to its full potential. For example, in two datasets we analyzed for cassette-driven divergent transcripts in strain backgrounds deficient for exosome-mediated RNA decay, cassette integrated mutants contained flanking loxP sites that were not utilized to excise the cassette (*Schmidt et al., 2012*). Use of a single-step 'insulated' cassette for genome editing maintains the efficiency of the originally designed system, while preventing the potentially catastrophic off-target effects that disruption of neighboring gene expression can cause.

While our study was limited to yeast, all the key factors behind the cassette-induced off-target effects described here are conserved in higher eukaryotes. Because of this, we hypothesize that genome-editing-driven transcriptional interference, including LUTI-based repression, is likely to occur in other organisms. It is now understood that promoters in yeast, mammals, and other eukaryotes are inherently bidirectional, but that evolution of local sequences to can limit transcription interference of adjacent genes by transcription termination of divergent transcripts within a short distance of their initiation (*Teodorovic et al., 2007*; *Neil et al., 2009*; *Seila et al., 2008*; *Churchman and Weissman, 2011*; *Jin et al., 2017*; *Schulz et al., 2013*; *Arnone, 2020*; *Villa et al., 2020*; *Ha et al., 2022*; *Ntini et al., 2013*; *Almada et al., 2013*; *Core et al., 2008*; *Xu et al., 2009*). Although NNS-based

termination does not prevent production of long stable divergent transcripts from bidirectional promoters in mammals, analogous evolution of early polyadenylation sites appears to have achieved the same result (*Schulz et al., 2013*; *Ntini et al., 2013*; *Almada et al., 2013*; *Porrua and Libri, 2015*). In both cases, the local context of promoters defines the degree to which they produce divergent transcripts with the ability to interfere with neighboring gene expression.

The bidirectional nature of commonly used expression cassette promoters, including *pTEF* in yeast, and the CMV, eEF1α, and SV40 promoters in mammalian systems, has long been known (*Curtin et al., 2008*; *Gidoni et al., 1985*). However, consequences of this for their modular use in expression constructs do not yet seem to be widely considered. Early studies that established 'minimal' promoters for modular use in expression cassettes explicitly served to separate the transcript-generating feature of promoter regions from their native sequence context. Because local sequence features outside of these minimal promoter regions can limit the effects of divergent transcription, and insertion of a terminator flanking inserted promoter sequences can serve a similar function, our study suggests this simple fix to prevent cassette-based off-target effects in future studies. Our study also emphasizes the need for proper controls to establish causality of observed phenotypes to the intended mutation, even when using precision genome engineering approaches in yeast and other organisms.

# Materials and methods

## Key resources table

| Reagent type (species) or resource | Designation | Source or reference | Identifiers | Additional information |
|---|---|---|---|---|
| Recombinant DNA reagent | pFA6a-kanMX6 | *Longtine et al., 1998* | pUB1 | pFA6a backbone expressing *kanr* (Geneticin resistance) from *TEF* promoter and terminator |
| Recombinant DNA reagent | pFA6a-natMX6 | *Longtine et al., 1998* | pUB153 | pFA6a backbone expressing *natr* (Nourseothricin resistance) from *TEF* promoter and terminator |
| Recombinant DNA reagent | pFA6a-his3MX | *Longtine et al., 1998* | pUB3 | pFA6a backbone expressing *S. pombe HIS5* (complements *S. cerevisiae his3*) from *TEF* promoter and terminator |
| Recombinant DNA reagent | pFA6a-TRP1 | *Longtine et al., 1998* | pUB2 | pFA6a backbone expressing *TRP1* from *TRP1* promoter and terminator |
| Recombinant DNA reagent | kanMX6-ins1 | This paper | pUB2255 | pFA6a backbone with *tDEG1* insulators flanking insert expressing KANr (Geneticin resistance) from TEF promoter and terminator (pFA6a-tDEG1-kanMX6-tDEG1) |
| Recombinant DNA reagent | kanMX6-ins2 | This paper | pUB2272 | pFA6a backbone with *tDEG1* and tCYC1 insulators flanking insert expressing kanr (Geneticin resistance) from TEF promoter and terminator (pFA6a-tDEG1-kanMX6-tCYC1) |
| Recombinant DNA reagent | TRP1MX6-ins4 | This paper | pUB2308 | pFA6a backbone with *tDEG1* and *tCYC1* insulators flanking insert expressing *TRP1* from *TEF* promoter and terminator (pFA6a-tDEG1-TRP1MX6-tCYC1) |
| Recombinant DNA reagent | his3MX6-ins3 | This paper | pUB2309 | pFA6a backbone with *tDEG1* and *tCYC1* insulators flanking insert expressing *S. pombe HIS5* (complements *S. cerevisiae his3*) from *TEF* promoter and terminator (pFA6a-tDEG1-HISMX6-tCYC1) |
| Recombinant DNA reagent | natMX6-ins5 | This paper | pUB2310 | pFA6a backbone with *tDEG1* and *tCYC1* insulators flanking insert expressing natr (Clonat resistance) from *TEF* promoter and terminator (pFA6a-tDEG1-NATMX6-tCYC1) |
| Recombinant DNA reagent | pDBP1-DBP1 | This paper | pUB1761 | *leu2::LEU2* single integration vector, expresses *DBP1* ORF from natural regulatory sequences (~1 kb upstream and ~1 kb downstream of ORF) |
| Recombinant DNA reagent | pDBP1-DBP1-3V5 | This paper | pUB1831 | *leu2::LEU2* single integration vector, expresses *DBP1* ORF tagged with 3v5 expressed from natural regulatory sequences (~1 kb upstream and ~1 kb downstream of ORF) |

*Continued on next page*

*Continued*

| Reagent type (species) or resource | Designation | Source or reference | Identifiers | Additional information |
|---|---|---|---|---|
| Recombinant DNA reagent | pMRP51-MRP51 | This paper | pUB2401 | *leu2::LEU2* single integration vector, expresses *MRP51* ORF from natural regulatory sequences (~0.5 kb upstream and ~0.5 kb downstream of ORF) |
| Recombinant DNA reagent | kanMX6-ins6 | This paper | pUB2434 | pFA6a backbone with *tDEG1* insulator flanking *TEF* promoter that expresses kanr (Geneticin resistance) (pFA6a-tDEG1-kanMX6) |
| Recombinant DNA reagent | natMX6-ins7 | This paper | pUB2435 | pFA6a backbone with *tCYC1* flanking *pPGK1* from *C. glabrata* which expresses natr and is terminated by *tPGK1* from *C. glabrata* (pFA6a-tCYC1-pPGK1-natR-tPGK1) |
| Recombinant DNA reagent | short-pTEF1-kanMX6 | This paper | | pFA6a backbone but with *S. cerevisiae* short *pTEF1* |
| Recombinant DNA reagent | long-pTEF1-kanMX6 | This paper | | pFA6a backbone but with *S. cerevisiae* long *pTEF1* |
| Recombinant DNA reagent | NNS-kanMX6 | This paper | | pFA6a backbone but with *SNR13* NNS termination site inserted 5' of the *pTEF* from *A. gossypii* |
| Sequence-based reagent | F deletion primer *DBP1* | This paper | 7009 | AAGGAGTTCTATATTTGGGTTACTCTT TTGTTCTTTCAGCgaattcgagctcgtttaaac |
| Sequence-based reagent | R deletion primer *DBP1* | This paper | 6678 | TAAAAAAAAAACCCTTTGAGTGAAAGT ATTACAAGAAAAACGGATCCCCGGGTTAATTAA |
| Strain, strain background (*Saccharomyces cerevisiae*) | Wild-type | ***Padmore et al., 1991*** UB13 | | MATa, ho::LYS2, lys2, ura3, leu2::hisG, his3::hisG, trp1::hisG (SK1 wild-type) |
| Strain, strain background (*Saccharomyces cerevisiae*) | Wild-type | ***Brar et al., 2012*** UB15 | | MATa/MATalpha, ho::LYS2/ho::LYS2, lys2/lys2, ura3/ura3, leu2::hisG/leu2::hisG, his3::hisG/his3::hisG, trp1::hisG/trp1::hisG (SK1 wild-type) |
| Strain, strain background (*Saccharomyces cerevisiae*) | dbp1Δ::kanMX6 | This paper | UB15798 | MATa/MATalpha, ho::LYS2/ho::LYS2, lys2/lys2, ura3/ura3, leu2::hisG/leu2::hisG, his3::hisG/his3::hisG, trp1::hisG/trp1::hisG, dbp1Δ::kanMX6/dbp1Δ::kanMX6 |
| Strain, strain background (*Saccharomyces cerevisiae*) | Wild-type, MRP51-3V5 | This paper | UB32228 | MATa, ho::LYS2, lys2, ura3, leu2::hisG, his3::hisG, trp1::hisG, MRP51-3V5 |
| Strain, strain background (*Saccharomyces cerevisiae*) | dbp1Δ::kanMX6, MRP51-3V5 | This paper | UB32229 | MATa, ho::LYS2, lys2, ura3, leu2::hisG, his3::hisG, trp1::hisG, dbp1Δ::kanMX6, MRP51-3V5 |
| Strain, strain background (*Saccharomyces cerevisiae*) | Wild-type | This paper | UB22843 | MATa/MATalpha, ho::LYS2/ho::LYS2, lys2/lys2, ura3/ura3, LEU2/LEU2, his3::hisG/his3::hisG, trp1::hisG/trp1::hisG, DED1::DED1-3V5::HIS3/DED1::DED1-3V5 |
| Strain, strain background (*Saccharomyces cerevisiae*) | dbp1Δ::kanMX6 | This paper | UB22958 | MATa/MATalpha, ho::LYS2/ho::LYS2, lys2/lys2, ura3/ura3, LEU2/LEU2, his3::hisG/his3::hisG, trp1::hisG/trp1::hisG, dbp1Δ::kanMX6/dbp1Δ::kanMX6, DED1::DED1-3V5::HIS3/DED1::DED1-3V5 |
| Strain, strain background (*Saccharomyces cerevisiae*) | dbp1Δ::kanMX6, leu2::pDBP1-DBP1-3V5::LEU2 | This paper | UB23951 | MATa/MATalpha, ho::LYS2/ho::LYS2, lys2/lys2, ura3/ura3, leu2::hisG/leu2::hisG, HIS3/HIS3, trp1::hisG/trp1::hisG, dbp1Δ::kanMX6/dbp1Δ::kanMX6, leu2::pDBP1-DBP1-3V5::LEU2/leu2::pDBP1-DBP1-3V5::LEU2 |
| Strain, strain background (*Saccharomyces cerevisiae*) | dbp1Δ::kanMX6, leu2::pMRP51-MRP51::LEU2 | This paper | UB34918 | MATa/MATalpha, ho::LYS2/ho::LYS2, lys2/lys2, ura3/ura3, leu2::hisG/leu2::hisG, his3::hisG/his3::hisG, trp1::hisG/trp1::hisG, dbp1Δ::kanMX6/dbp1Δ::kanMX6, leu2::pMRP51-MRP51::LEU2/leu2::pMRP51-MRP51::LEU2 |

*Continued on next page*

*Continued*

| Reagent type (species) or resource | Designation | Source or reference | Identifiers | Additional information |
|---|---|---|---|---|
| Strain, strain background (*Saccharomyces cerevisiae*) | *dbp1Δ::kanMX6, leu2::pDBP1-DBP1::LEU2* | This paper | UB24206 | *MATa/MATalpha, ho::LYS2/ho::LYS2, lys2/lys2, ura3/ura3, leu2::hisG/leu2::hisG, HIS3/HIS3, trp1::hisG/trp1::hisG, dbp1Δ::kanMX6/dbp1Δ::kanMX6, leu2::pDBP1-DBP1::LEU2/leu2::pDBP1-DBP1::LEU2* |
| Strain, strain background (*Saccharomyces cerevisiae*) | *dbp1Δ::kanMX6* | This paper | UB15008 | *MATa, ho::LYS2, lys2, ura3, leu2::hisG, his3::hisG, trp1::hisG, dbp1Δ::kanMX6* |
| Strain, strain background (*Saccharomyces cerevisiae*) | *dbp1Δ::natMX6* | This paper | UB24358 | *MATa, ho::LYS2, lys2, ura3, leu2::hisG, his3::hisG, trp1::hisG, dbp1Δ::natMX6* |
| Strain, strain background (*Saccharomyces cerevisiae*) | *dbp1Δ::his3MX* | This paper | UB24360 | *MATa, ho::LYS2, lys2, ura3, leu2::hisG, his3::hisG, trp1::hisG, dbp1Δ::his3MX* |
| Strain, strain background (*Saccharomyces cerevisiae*) | *dbp1Δ::TRP1* | This paper | UB24362 | *MATa, ho::LYS2, lys2, ura3, leu2::hisG, his3::hisG, trp1::hisG, dbp1Δ::TRP1* |
| Strain, strain background (*Saccharomyces cerevisiae*) | *upf1Δ::natMX6* | This paper | UB25148 | *MATa/MATalpha, ho::LYS2/ho::LYS2, lys2/lys2, ura3/ura3, leu2::hisG/leu2::hisG, his3::hisG/his3::hisG, trp1::hisG/trp1::hisG, upf1Δ::natMX6/upf1Δ::natMX6* |
| Strain, strain background (*Saccharomyces cerevisiae*) | Wild type | This paper | UB24955 | *MATa/MATalpha, ho::LYS2/ho::LYS2, lys2/lys2, ura3/ura3, LEU2/LEU2, his3::hisG/his3::hisG, trp1::hisG/trp1::hisG* |
| Strain, strain background (*Saccharomyces cerevisiae*) | *dbp1Δ* | This paper | UB24950 | *MATa/MATalpha, ho::LYS2/ho::LYS2, lys2/lys2, ura3/ura3, LEU2/LEU2, his3::hisG/his3::hisG, trp1::hisG/trp1::hisG, dbp1Δ/dbp1Δ* |
| Strain, strain background (*Saccharomyces cerevisiae*) | *dbp1Δ::tDEG1-kanMX6-tDEG1, MRP51-3V5* | This paper | UB32497 | *MATa, ho::LYS2, lys2, ura3, leu2::hisG, his3::hisG, trp1::hisG, dbp1Δ::tDEG1-kanMX6-tDEG1, MRP51-3V5* |
| Strain, strain background (*Saccharomyces cerevisiae*) | *dbp1Δ::tDEG1-kanMX6-tCYC1, MRP51-3V5* | This paper | UB32498 | *MATa, ho::LYS2, lys2, ura3, leu2::hisG, his3::hisG, trp1::hisG, dbp1Δ::tDEG1-kanMX6-tCYC1, MRP51-3V5* |
| Strain, strain background (*Saccharomyces cerevisiae*) | *dbp1Δ::rev-kanMX6* | This paper | UB35165 | *MATa, ho::LYS2, lys2, ura3, leu2::hisG, his3::hisG, trp1::hisG, dbp1Δ::rev-kanMX6* |
| Strain, strain background (*Saccharomyces cerevisiae*) | *dbp1Δ::kanMX6-ins6* | This paper | N/A | *MATa, ho::LYS2, lys2, ura3, leu2::hisG, his3::hisG, trp1::hisG, dbp1Δ::kanMX6-ins6* |
| Strain, strain background (*Saccharomyces cerevisiae*) | *dbp1Δ::rev-kanMX6-ins6* | This paper | N/A | *MATa, ho::LYS2, lys2, ura3, leu2::hisG, his3::hisG, trp1::hisG, dbp1Δ::rev-kanMX6-ins6* |
| Strain, strain background (*Saccharomyces cerevisiae*) | *dbp1Δ::natMX6-ins7* | This paper | N/A | *MATa, ho::LYS2, lys2, ura3, leu2::hisG, his3::hisG, trp1::hisG, dbp1Δ::NATMX6-ins7* |
| Strain, strain background (*Saccharomyces cerevisiae*) | *dbp1Δ::rev-natMX6-ins7* | This paper | N/A | *MATa, ho::LYS2, lys2, ura3, leu2::hisG, his3::hisG, trp1::hisG, dbp1Δ::rev-NATMX6-ins7* |
| Strain, strain background (*Saccharomyces cerevisiae*) | *dbp1Δ::short-pTEF1-kanMX6* | This paper | UB35339 | *MATa, ho::LYS2, lys2, ura3, leu2::hisG, his3::hisG, trp1::hisG, dbp1Δ::short-pTEF1-kanMX6* |
| Strain, strain background (*Saccharomyces cerevisiae*) | *dbp1Δ::long-pTEF1-kanMX6* | This paper | UB35337 | *MATa, ho::LYS2, lys2, ura3, leu2::hisG, his3::hisG, trp1::hisG, dbp1Δ::long-pTEF1-kanMX6* |
| Strain, strain background (*Saccharomyces cerevisiae*) | *dbp1Δ::NNS-kanMX6* | This paper | UB35331 | *MATa, ho::LYS2, lys2, ura3, leu2::hisG, his3::hisG, trp1::hisG, dbp1Δ::NNS-kanMX6* |

## Strains

All strains were derived from the *Saccharomyces cerevisiae* SK1 background (*Padmore et al., 1991*). Detailed strain genotypes can be found in Key resources table. Strains were diploid *MATa/alpha* unless otherwise specified as haploid (*MATa*). Selection-mediated loss of function mutants were created by transforming wild-type haploids with a linear PCR fragment encoding a selection cassette that would replace the ORF of interest via homologous DNA repair mechanisms (*Longtine et al., 1998*; *Wach et al., 1997*; *Wach et al., 1994*; *Baudin et al., 1993*; *Lorenz et al., 1995*). Sequences directly upstream and downstream of each replaced ORF were always left intact. Genotypes for all strains were confirmed using PCR and mutants were backcrossed to a wild-type strain of the opposite mating type to ensure clearance of possible background mutations from transformation. The

marker-less *dbp1Δ* was created by transforming a wild-type haploid with a plasmid-encoding Cas9 and a sgRNA targeting the *DBP1* ORF, alongside a linear repair fragment that removes the *DBP1* ORF while preserving the upstream and downstream adjacent sequence. Strains containing the Mrp51-3v5 allele were created by transforming wild-type or the selection replaced *dbp1Δ* strain of interest with a plasmid-encoding Cas9 and a sgRNA targeting *MRP51*, alongside a linear repair template that carboxy-terminally tags Mrp51 with the 3v5 sequence and introduces a synonomous mutation in the guide target sequence to prevent re-editing. Strains expressing exogenous Dbp1 or Mrp51 from the *LEU2* locus were created by cloning the ORF of interest as well as ~1 kb of upstream and downstream regulatory sequence (for *DBP1*), or 500 bp of upstream and downstream regulatory sequence (for *MRP51*) into a single integration plasmid targeted to *LEU2.* This plasmid was linearized by digestion with SwaI nuclease (NEB, Ipswich, MA) and transformed into the *dbp1Δ::kanMX6* strain.

## Yeast growth conditions

Yeast were grown for experiments as in *Powers and Brar, 2021*. Briefly, strains were thawed on YPG plates from glycerol stocks overnight, then grown on YPD plates for a day before use in experiments. All growth experiments were carried out at 30°C using YEP media supplemented with 2% (wt/vol) of the carbon source, except for glycerol which was used at 3% (vol/vol). All growth experiments shown were repeated at least twice with one representative growth curve shown for each. Synchronous sporulating cells were prepared by first growing cells in YEP 2% dextrose for 24 hr at room temperature then diluted into BYTA at $OD_{600}$ = 0.25 and grown overnight at 30°C. Next, cells were pelleted, washed with water, and resuspended in sporulation (SPO) medium: either (0.5% KAc [pH = 7.0] supplemented with 0.02% raffinose) for *Figure 1* ribosome profiling and mRNA-seq experiments, or rich SPO medium (2% KAc [pH = 7.0] supplemented with 40 mg/l adenine), 40 mg/l uracil, 20 mg/l histidine, 20 mg/l leucine, 20 mg/l tryptophan for sporulation efficiency, polysome experiments, and ribosome profiling in *Figure 5—figure supplement 2*. Sporulation efficiency was counted at 24 hr after induction into SPO media and 200 cells were counted per sample for three independent biological replicates. Samples were blinded prior to counting.

## Ribosome profiling and mRNA-seq

Meiotic yeast cells were harvested after 4 or 6 hr in sporulation medium (*Figure 1* data: 4 hr; *Figure 3—figure supplement 1*: 6 hr), by brief treatment with 100 µg/ml cycloheximide then flash freezing. They were subjected to ribosome profiling and mRNA-seq as described in *Powers and Brar, 2021*. Reads per ORF were determined following mapping of all reads to the SK1 genome. RPKM values for ribosome footprints (FP) and mRNA reads were calculated by dividing the number of raw reads per gene by the total number of million mapped reads per sample and by the length in kilobases of each gene. TE was calculated by dividing the FP RPKM/mRNA RPKM for each gene. Data visualization of genome tracts was made using MochiView (*Homann and Johnson, 2010*).

## Immunoblotting

Trichloroacetic acid (TCA) extractions were performed to collect total protein from samples as described in *Chen et al., 2017*; *Chen et al., 2017*. Briefly, ~2.5 $OD_{600}$ of yeast were treated with 5% TCA at 4°C overnight, washed with acetone, dried then lysed in lysis buffer (50 mM Tris–HCl [pH 7.5], 1 mM ethylenediaminetetraacetic acid (EDTA)), 2.75 mM dithiothreitol (DTT), protease inhibitor cocktail (cOmplete EDTA-free, Roche) for 5 min on a Mini-Beadbeater-96 (Biospec Products). Next, 3× sodium dodecyl sulfate (SDS) sample buffer (187.5 mM Tris [pH 6.8], 6% β-mercaptoethanol, 30% glycerol, 9% SDS, 0.05% bromophenol blue) was added to 1× and the cell lysate was boiled for 5 min. Proteins were run on 4–12% Bis-Tris Bolt gels (Thermo Fisher) then transferred to 0.45 µM nitrocellulose membranes using the 30 min mixed molecular weight protocol on the Trans-Blot Turbo System from Bio-Rad. Membranes were blocked with Intercept (phosphate-buffered saline, PBS) blocking buffer (LI-COR Biosciences, Lincoln, NE) at room temperature then incubated overnight at 4°C with mouse anti-v5 (1:1000, Invitrogen, RRID:AB_2556564) and rat anti-tubulin alpha (1:10,000, Serotec, RRID:AB_325005). Membranes were washed in PBS-0.08% Tween then incubated with an anti-mouse secondary antibody conjugated to IR Dye 800 (RRID:AB_621842) at a 1:15,000 dilution, and either an anti-rat secondary antibody conjugated to IR Dye 680 (RRID:AB_10956590) at a 1:15,000 dilution at

room temperature for 1–2 hr (LI-COR Biosciences). Immunoblot images were generated and quantified using the Odyssey system (LI-COR Biosciences).

## Polysome analysis

Cells were treated with 100 µg/ml cycloheximide for 30 s then filter collected, flash frozen, lysed, and prepared for sucrose gradient centrifugation of as in *Hughes et al., 2019*, with the following exceptions. Samples were thawed just prior to their loading on sucrose gradients, and SUPERase·In was added as for 'mock digested samples'. Also, the polysome lysis buffer used was modified to contain protease and phosphatase inhibitors and is as follows: 20 mM Tris pH 8, 140 mM KCl, 1.5 mM MgCl$_2$, 100 µg/ml cycloheximide, 1% Triton X-100, 2 µg/ml Aprotinin, 10 µg/ml Leupeptin, 1 mM phenylmethylsulfonyl flouride (PMSF), 1:100 PIC2, and 1:100 PIC3. Lysates were loaded onto 7–47% sucrose gradients and spun in a SW 41 Ti rotor for 3 hr at 35,000 rcf at 4°C. Following the spin, samples were kept at 4°C and immediately collected using the Gradient Master gradient station from BioComp while monitoring 260 nm absorbance with the Bio-Rad EM-1 Economonitor. Samples were fractionated as shown in *Figure 2*, flash frozen, and submitted for mass spectrometry analysis.

## Polysome fraction processing for LC–MS/MS measurements

Proteins were precipitated and desalted using the SP3 method for liquid chomatography with tandem mass spectrometry (LC-MS/MS) as described in *Hughes et al., 2019*; *Cox and Mann, 2008*. 50% from each polysome fraction (volume) were processed. Disulfide bonds were reduced with 5 mM dithiothreitol and cysteines were subsequently alkylated with 10 mM iodoacetamide. Proteins were precipitated on 0.5 µg/µl speedBead magnetic carboxylated modified beads (1:1 mix of hydrophobic and hydrophilic beads, cat# 6515215050250, 45152105050250, GE) by addition of 100% ethanol in a 1:1 (vol:vol) sample:ethanol ratio followed by 15 min incubation at 25°C, 1000 rpm. Protein-bound beads were washed in 80% ethanol and proteins were digested off the beads by addition of 0.8 µg sequencing grade modified trypsin (Promega) in 100 mM ammonium bicarbonate, incubated 16 hr at 25°C, 600 rpm. Beads were removed and the resulting tryptic peptides evaporated to dryness in a vacuum concentrator. Dried peptides were further desalted by another round of SP3 precipitation; peptides were reconstituted in 200 µl of 95% acetonitrile (ACN), followed by addition of bead-mix to a final concentration of 0.5 µg/µl and incubated 15 min at 25 °C, 1000 rpm. Beads were subsequently washed in 80% ethanol, peptides were eluted off the beads in 50 µl 2% dimethyl sulfoxide and samples were evaporated to dryness in a vacuum concentrator. Dried peptides were then reconstituted in 3% ACN/0.2% formic acid to a final concentration of 0.5 µg/µl.

## LC–MS/MS analysis on a Q-Exactive HF

About 1 µg of total peptides were analyzed on a Waters M-Class UPLC using a 25 cm Ionopticks Aurora column coupled to a benchtop Thermo Fisher Scientific Orbitrap Q Exactive HF mass spectrometer. Peptides were separated at a flow rate of 400 nl/min with a 190-min gradient, including sample loading and column equilibration times. Data were acquired in data-dependent mode using Xcalibur software. MS1 Spectra were measured with a resolution of 120,000, an AGC target of 3e6 and a mass range from 300 to 1800 *m/z*. Up to 12 MS2 spectra per duty cycle were triggered at a resolution of 15,000, an AGC target of 1e5, an isolation window of 1.6 *m/z* and a normalized collision energy of 27.

All raw data were analyzed with MaxQuant software version 1.6.10.43 (*de Hoon et al., 2004*) using a UniProt yeast database (release 2014_09, strain ATCC 204508/S288c), and MS/MS searches were performed with the following parameters: oxidation of methionine and protein N-terminal acetylation as variable modifications; carbamidomethylation as fixed modification; trypsin/P as the digestion enzyme; precursor ion mass tolerances of 20 ppm for the first search (used for nonlinear mass recalibration) and 4.5 ppm for the main search, and a fragment ion mass tolerance of 20 ppm. For identification, we applied a maximum false discovery rate of 1% separately on protein and peptide level. 'Match between the runs' was activated, as well as the 'LFQ' (label-free quantification) normalization (at least two ratio counts were necessary to get an LFQ value). We required 1 or more unique/razor peptides for protein identification and a ratio count of 2 or more for label-free protein quantification in each sample. This gave intensity values for a total of 2143 protein groups across both replicates. 'LFQ' normalized values were used for all subsequent analyses.

LFQ normalized values were then clustered with Cluster 3.0 (*Saldanha, 2004*) and visualized with Java TreeView (*Song et al., 2016*). Proteins not quantified in 2 or more samples were excluded in clustering. GO enrichment was assessed on the clusters specified (June 2022) and compared to the background population of all proteins quantified in the experiment (*Boyle et al., 2004*).

### Radioactive amino acid incorporation assays

Cells were transferred to sporulation (SPO) media and incubated at 30°C for 4 hr. To metabolically label the cells 5 µL of EasyTag EXPRESS $^{35}$S protein labeling mix (PerkinElmer), was added to 10 ml of SPO cultures and incubated with shaking for 10 min at 30°C. Protein was precipitated by addition of 100 µl 100% TCA to 900 µl SPO culture and incubated at 95°C with shaking. Samples were chilled on ice then pelleted and washed with cold 10% TCA, followed by a wash in cold 100% ethanol. Samples were resuspended in 5 ml of Econo-Safe scintillation fluid (RPI). Scintillation was counted for 2 min and the $^{35}$S incorporation rates were derived from counts per minute normalized to cell density between samples and to wild-type measurements.

### Design and testing of insulated selection cassettes

Based on the previous report that the *DEG1* and *CYC1* terminators can be used as insulator sequences (*Song et al., 2016*), we amplified these terminators from wild-type yeast and placed them into the 5′ and 3′ ends of the pFA6a-kanMX6 backbone using gibson assembly. Insulators were placed just inside the standard F1 and R1 primer amplification sites such that they would be useable with any primers designed to the previous system (pFA6a; *Longtine et al., 1998*). Insulated plasmids were fully sequenced over cloning junctions and the entire region to be amplified and integrated into yeast during transformation. To test the ability of these insulators to prevent aberrant transcription at the *DBP1* locus, the kanMX6-ins cassettes were amplified and transformed into yeast exactly as had been done for the previous kanMX6 selection cassettes used in this study. When used to replace *DBP1* ORF, Mrp51-3v5 levels and 5′ RACE confirmed the efficacy of the kanMX6-ins cassettes in preventing aberrant LUTI regulation of *MRP51*. We then replaced kan$^r$ in the insulated plasmids with three additional markers for selection (including *S. pombe HIS5* amplified from pFA6a-HIS3MX6): complements *S. cerevisiae* his3, *TRP1* amplified from pFA6a-TRP1, and nat$^r$ such that they can be used conveniently in combination as was done for the original toolkit (*Longtine et al., 1998*; *Wach et al., 1994*). To clone kanMX6-ins6 primers were used to amplify the entire kanMX6-ins2 except for the 3′ tCYC1 terminator and the Q5 Site-Directed Mutagenesis Kit was used to circularize the plasmid. For natMX6-ins7 a gBlocks Gene Fragment containing the *PGK1* promoter and terminator from *C. glabrata* expressing nat$^r$ and 5′ flanked by the *S. cerevisiae CYC1* terminator was cloned into the pFA6a backbone. Detailed descriptions of these plasmids and can be found in Key resources table.

### 5′ RACE

5′ RACE was performed with total RNA extracted from samples using phenol–chloroform precipitation and the GeneRacer Kit with SuperScript III RT and Topo TA Cloning Kit for Sequencing. For each sample, 4.8 µg of total RNA was used. Random primers were used to reverse transcribe the decapped cDNA library. The 5′ ends of the *MRP51* cDNA were then amplified using the GeneRacer 5′ primer and a reverse gene-specific primer (5′ ACCGTCCAAGCAACTCTGCCAATGTC 3′) in a reaction with Platinum *Taq* High Fidelity DNA Polymerase. Amplified 5′ ends were then cloned into a Topo TA Cloning vector and clones were miniprepped and sequenced using Sanger sequencing. SnapGene was used to visualize and align the sequenced cDNA ends to the corresponding genomic position.

### Materials availability

All newly created materials are available upon request. Newly created 'Insulated' selection cassettes will also be deposited to Addgene and made publicly available.

## Acknowledgements

We thank Nick Ingolia, Elçin Ünal, and current members of the Brar and Ünal labs for their comments on this manuscript. We thank Calvin Jan, Stephen Floor, Nick Ingolia, James Olzmann, and James Nuñez for sharing perspectives on mammalian genome editing, and David Brow for insight into NNS-mediated termination. We thank Guillaume Chanfreau and Samuel Demario for generously sharing

data. This work was supported by National Institutes of Health funding to GAB (R35GM134886) and MJ (R35GM128802). ENP was funded by an NSF predoctoral Fellowship.

## Additional information

### Funding

| Funder | Grant reference number | Author |
|---|---|---|
| National Institutes of Health | R35GM134886 | Gloria Ann Brar |
| National Institutes of Health | R35GM128802 | Marko Jovanovic |
| National Science Foundation | | Emily Nicole Powers |

The funders had no role in study design, data collection, and interpretation, or the decision to submit the work for publication.

### Author contributions

Emily Nicole Powers, Conceptualization, Data curation, Formal analysis, Supervision, Funding acquisition, Validation, Investigation, Visualization, Writing – original draft, Project administration, Writing – review and editing; Charlene Chan, Lidia Llacsahuanga Allcca, Jenny Kim Kim, Investigation; Ella Doron-Mandel, Data curation, Investigation, Writing – review and editing; Marko Jovanovic, Data curation, Methodology, Project administration; Gloria Ann Brar, Conceptualization, Data curation, Formal analysis, Supervision, Funding acquisition, Validation, Investigation, Visualization, Methodology, Writing – original draft, Project administration, Writing – review and editing

### Author ORCIDs

Emily Nicole Powers (iD) http://orcid.org/0000-0003-4497-7318
Gloria Ann Brar (iD) http://orcid.org/0000-0002-8560-9581

### Decision letter and Author response

Decision letter https://doi.org/10.7554/eLife.81086.sa1
Author response https://doi.org/10.7554/eLife.81086.sa2

## Additional files

### Supplementary files
• MDAR checklist

### Data availability

mRNA sequencing and ribosome profiling data are available at NCBI GEO, with accession numbers GSE207267 and GSE207189. Mass spectrometry data are available at MassIVE, with accession number MSV000089724.

The following datasets were generated:

| Author(s) | Year | Dataset title | Dataset URL | Database and Identifier |
|---|---|---|---|---|
| Powers E, Brar G | 2022 | Loss of *DBP1* during meiosis in yeast | https://www.ncbi.nlm.nih.gov/geo/query/acc.cgi?acc=GSE207267 | NCBI Gene Expression Omnibus, GSE207267 |

*Continued on next page*

*Continued*

| Author(s) | Year | Dataset title | Dataset URL | Database and Identifier |
|---|---|---|---|---|
| Powers E, Brar G | 2022 | Ribosome profiling and mRNA seq demonstrate that insertion of common resistance cassettes in yeast drives aberrant transcription | https://www.ncbi.nlm.nih.gov/geo/query/acc.cgi?acc=GSE207189 | NCBI Gene Expression Omnibus, GSE207189 |
| Jovanovic M, Kim Kim J | 2022 | Use of a ubiquitous gene-editing tool in budding yeast causes off-target repression of neighboring gene protein synthesis | https://massive.ucsd.edu/ProteoSAFe/dataset.jsp?task=972212bec77b48faab383a6c5c2b9310 | MSV000089724, MassIVE |

The following previously published datasets were used:

| Author(s) | Year | Dataset title | Dataset URL | Database and Identifier |
|---|---|---|---|---|
| Brar GA, Cheng Z | 2018 | Small and large ribosomal subunit deficiencies lead to distinct gene expression signatures that reflect cellular growth rate | https://www.ncbi.nlm.nih.gov/geo/query/acc.cgi?acc=GSE121189 | NCBI Gene Expression Omnibus, GSE121189 |
| Sen ND, Hinnebusch AG | 2018 | *DBP1* in budding yeast | https://www.ncbi.nlm.nih.gov/geo/query/acc.cgi?acc=GSE111255 | NCBI Gene Expression Omnibus, GSE111255 |

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
