## [Editor Report]

The power of yeast genetics frequently depends on the insertion of selectable expression cassettes. The authors demonstrate that an unfortunate vulnerability of these cassettes lies in the inevitable divergent antisense transcription that is produced, which can suppress the expression of proximal genes. The authors provide mechanistic insight into this consequence of yeast genomic editing and provide solutions that can be used for all such cassettes.

---

## [Decision Letter]

**Decision letter after peer review:**

[Editors’ note: the authors submitted for reconsideration following the decision after peer review. What follows is the decision letter after the first round of review.]

Thank you for submitting the paper "Use of a ubiquitous gene-editing tool in budding yeast causes off-target repression of neighboring gene protein synthesis" for consideration by *eLife*. Your article has been reviewed by 2 peer reviewers, and the evaluation has been overseen by a Reviewing Editor and a Senior Editor. The reviewers have opted to remain anonymous.

Comments to the Authors:

We are sorry to say that, after consultation with the reviewers, we have decided that this work will not be considered further for publication by *eLife*, because the conclusions that can be drawn from the work are not general enough for the broad readership of *eLife*. We hope that the reviewers' comments and suggestions will be helpful for submission to another journal.

*Reviewer #1 (Recommendations for the authors):*

1. Line 82 – It should be noted that the deletion set strains have long been shown to be imperfect, as there are many types of genetic modifications within the collection. The relevant references are:

Widespread aneuploidy revealed by DNA microarray expression profiling.

Hughes TR, Roberts CJ, Dai H, Jones AR, Meyer MR, Slade D, Burchard J, Dow S, Ward TR, Kidd MJ, Friend SH, Marton MJ. Nat Genet. 2000 Jul;25(3):333-7. doi: 10.1038/77116.

Genome-wide consequences of deleting any single gene.

Teng X, Dayhoff-Brannigan M, Cheng WC, Gilbert CE, Sing CN, Diny NL, Wheelan SJ, Dunham MJ, Boeke JD, Pineda FJ, Hardwick JM. Mol Cell. 2013 Nov 21;52(4):485-94. doi: 10.1016/j.molcel.2013.09.026. Epub 2013 Nov 7.

2. Lines 83-85 – Seamless gene replacement has been available in yeast long before CRISPR/Cas9-based editing became available. One very common way is described in this reference:

Two-step method for constructing unmarked insertions, deletions and allele substitutions in the yeast genome. Gray M, Piccirillo S, Honigberg SM. FEMS Microbiol Lett. 2005 Jul 1;248(1):31-6. doi: 10.1016/j.femsle.2005.05.018.

3. Lines 269-271 – Was it experimentally verified that the ectopic copy of DBP2 was expressed at normal levels?

4. For all of the growth curves shown in Figure 2 and its supplement, we would be better able to assess the growth if the Y axis was a log scale since the cells are presumably growing exponentially. In addition, it's possible that the cell size or clumpiness of the mutants is different compared to wild type in a way that might affect the OD600 reading.

5. Figure 2 and supplement – The authors present convincing data that the mutant phenotypes observed in their dbp2 knockouts are not complemented by the integration of an exogenous copy of DBP2. However, in some sense, this is negative data, the failure to rescue the mutant phenotype. There are some additional strains whose analysis would be informative. Most obviously, if the dbp1 deletion phenotypes are really caused by reduced expression of MRP51, then integration of a second copy of MRP51 would be very strongly predicted to rescue the dpb1 deletion mutant phenotypes. Second, one would expect that the dpb1 and mrp51 deletions would have similar phenotypes, depending on how much residual Mrp51 remains in the dpb1 deletion.

6. For the experiment in Figure 2F, were opposite orientations examined for these cassettes? From what is written in lines 400-402 it sounds like integration of the standard resistance cassettes in the opposite orientation would abolish the off-target effect in at least some of these cases. Is that known or predicted?

7. line 461 -"Ablate" implies complete loss of function, which isn't the case for the examples shown. Use of a different word, such as "reduce" or "impair" would be more accurate. The severity of the effect would be better understood by comparison of the dbp1 and mrp51 deletions.

8. In the text the authors give their reconfigured cassette a simple and clear name – KANMX-ins. Therefore, for Figure 4 and its supplement, it would be clearer if the "KANMX-ins" name was used for the labels instead of the more complex name that is used.

The comments above should be addressed by experimental or writing changes. The key experiment that should be included pertains to comment 5 above. That will test whether the integration of an ectopic copy of MRP51 rescues the dpb1 deletion phenotype and whether the dpb1 deletion phenotype mimics that of an mrp51 deletion. Both of these would greatly fortify the conclusion that the dpb1 phenotype is caused by reduced mrp51 expression.

*Reviewer #2 (Recommendations for the authors):*

Suggestions for the authors for the presentation of their study.

General: if possible, please make clear at the level of individual figures and legends that all presented observations are reproducible. Examples would be if growth curves are n=2, potentially show both or show average and range, and state in figure legend instead of methods; for western blot quantification please indicate what the quantification represents – a single representative of experiment done more than once or an average of multiple experiments? If more than one, please add an error or range to the values. If one but representative, please state explicitly.

1. Line 191. "Together these data indicate that cassette replacement of the DBP1 ORF drives aberrant transcription independent of strain, cassette selection identity, and experimental conditions. Instead, the observed misregulation of MRP51 appears to be a general side-effect of the cassette-insertion."

This statement is too strong. In the experiment shown, two marker cassettes that may or may not be related are tested, apparently in the identical orientation. It is not clear if hygMX4 is the standard name for this marker. Is what is meant hphMX4? hphMX4 and kanMX6 will both use TEF promoter and terminator and in the case here, it appears that both markers were used in the same orientation, which as we learn later is likely critical for the mechanism, though not explicitly tested. This means that conclusion that the effect is general for cassette insertion has not been demonstrated, and potentially would not even be expected (based on the understood mechanism).

2. In Several places "believe" is used. Consider "propose" "argue" "hypothesize" "favor" in place.

3. line 296 "Insertion of every resistance cassette tested lead to a similar growth defect in a non-fermentable carbon source (Figure 2F), a phenotype which was not dependent on Dbp1 (Figure 2B)."

In the manuscript, as currently written, the concept of bidirectional transcription directed from the marker promoter is left to mention later. Since the tested markers share attributes, the bidirectional model is an attractive explanation at this position in the manuscript and a straightforward test of the proposed mechanism would be to use the markers in the opposite orientation. This experiment was not done but could be done. It has the added value of directly demonstrating that the effect is not innate to the marker but to the specific context in use.

4. For figure 2A, an additional panel with a difference map of some kind between the two strains could have value in visually demonstrating that the selected groups of genes are or are not the only major groups affected. There appear to be other clusters also affected and a description of these could add value to the presentation – for example, if it were discussed what is known about cellular phenotypes when mitochondrial translation is perturbed in a manner similar to reduced expression of components and how these data merely recapitulate previous observations or go beyond to describe impacts on the proteome that have not been described.

5. For figure 2f, the lack of complementation is a negative result because it is not demonstrated that DBP1 at the ectopic locus is transcribed and translated to the same extent as at the native locus. This is a minor concern because of the use of insulated KO cassette later but examination of the described CRISPR allele, or reversed kanMX6 or hphMX4 cassettes, or a CRISPR allele removing the DPB1 promoter and TSS, or introduction of a stop codon early in the ORF would have been a nice addition to this experiment.

6. Figure 2- fs 1A. The WT sample presumably has an error in the measurement. Even though mutant and WT are normalized there should be a way to propagate the error to the WT to indicate it is 100% +/- SD.

7. Line 357. "For the last case, in which UPF1 was replaced with the NATMX6 cassette, an abundant aberrant transcript was observed." While it is noted later, it would be appropriate here to note that upf1∆ itself may alter the stability of cryptic transcripts and this could be the reason for the especially strong transcript seen here.

8. Line 370 "Seamless deletion of the DBP1 ORF by this strategy led to the production of a transcript containing only the 5' and 3' UTRs of DBP1. Expression of this mutant transcript led to the aberrant translation of a dubious ORF contained within the DBP1 3' UTR, not translated in wild type (Figure 3—figure supplement 1).

It is a useful point to make that any genetic alteration may have unintended consequences (and this is a nice additional example) but the wording seems to unnecessarily conflate CRISPR-mediated editing with one specific time of deletion (removal of ORF) instead of the many different ways gene function may be rendered null using CRISPR editing. The statement just above states "effects" on "surrounding genes" plural but is not clear outside of the 3 UTR ORF as MRP51 shows a 1.2X effect- is this latter effect significant? If not, what other genes are affected? It does not seem necessary to use this example as justification for the new generation of selectable markers that are introduced in the next section- their value stands on their own, regardless of the extra effects of particular types of CRISPR editing.

9. Figure 3. An alternative framing would be that selectable markers employing the TEF1 promoter can drive or induce transcription divergently from the cassette in multiple contexts.

10. Line 492 "These features included a gene antisense and 5' to the cassette.."

This could be more accurately described as KO cassettes can have two orientations relative to the knocked out genes and adjacent genes are not technically antisense unless they occupy the same sequence on the other strand. Consider "These features include an upstream gene transcribed divergently from that of the orientation of the selectable marker gene in the cassette".

11. One consideration that would greatly improve the value of the new selectable markers created would be a design strategy that diversifies insulator sequences such that cassettes would not share flanking sequences. One issue with the kanMX6/hphMX4/natMX etc cassettes is that they share TEF promoter and terminator. This means that using them in the same strain reduces the efficiency of use by about 10-fold, due to marker exchange instead of integration into the desired target when one cassette is already in use in a strain. Having additional same sequences on each end can exacerbate this. One possibility would be to switch 5' and 3' positions but another would be to deploy at least two additional terminator regions to reduce marker exchange propensity.

12. Line 608 "c terminally" → "C-terminally" or "carboxy-terminally".

13. Line 634 "100 µM" of what?

14. Line 652 Fischer → Fisher.

15. Line 751 "We then cloned 3 additional cassettes using varying selection (including Saccharomyce *S. pombe* HIS5: complements S.cerevisiae HIS3, TRP1, and NATr)…"

Please make clear in methods if these markers were used to replace kan in kanMX6 or if insulators were added to those original cassettes and cite the source of the original cassettes (plasmids) and their official cassette names here for the relevant markers.

16. Line 756. 5' RACE. The nature of the sequencing here is not described. Numbers of "reads" are described, but is it instead meant numbers of clones with cDNA ends mapping to indicated positions because that is my understanding of how the GeneRacer III TOPO kit is designed to be employed. Additionally, were RACE products attempted to be mapped within the cassette and not found, or was it not examined? Are the new products at MRP51 within or external to the cassette and how close to the junction? It would be useful to know the distance between the cassette TSSs for the resistance marker and the induced divergent transcription to get an idea of if the properties are consistent with divergent transcription as described for single nucleosome-free regions or if instead the TEF one sequence used in these cassettes actually has an additional promoter region attached

---

## [Author Response]

[Editors’ note: The authors appealed the original decision. What follows is the authors’ response to the first round of review.]

Reviewer #1 (Recommendations for the authors):1. Line 82 – It should be noted that the deletion set strains have long been shown to be imperfect, as there are many types of genetic modifications within the collection. The relevant references are:Widespread aneuploidy revealed by DNA microarray expression profiling.Hughes TR, Roberts CJ, Dai H, Jones AR, Meyer MR, Slade D, Burchard J, Dow S, Ward TR, Kidd MJ, Friend SH, Marton MJ. Nat Genet. 2000 Jul;25(3):333-7. doi: 10.1038/77116.Genome-wide consequences of deleting any single gene.Teng X, Dayhoff-Brannigan M, Cheng WC, Gilbert CE, Sing CN, Diny NL, Wheelan SJ, Dunham MJ, Boeke JD, Pineda FJ, Hardwick JM. Mol Cell. 2013 Nov 21;52(4):485-94. doi: 10.1016/j.molcel.2013.09.026. Epub 2013 Nov 7.

We agree and have included these references in the revised manuscript. These reported issues with the deletion collection have helped researchers better guide their interpretation of deletion collection data, and in some cases correct errors in publications that are based on these artifacts. However, the deletion collection and data derived from it is still useful, and widely used. The off-target effect that we report here (a divergent, repressive transcript driven from *pTEF* when inserted at a non-endogenous locus*)* is not currently being widely considered as researchers design and interpret either small-scale or deletion-collection-based experiments.

2. Lines 83-85 – Seamless gene replacement has been available in yeast long before CRISPR/Cas9-based editing became available. One very common way is described in this reference:Two-step method for constructing unmarked insertions, deletions and allele substitutions in the yeast genome. Gray M, Piccirillo S, Honigberg SM. FEMS Microbiol Lett. 2005 Jul 1;248(1):31-6. doi: 10.1016/j.femsle.2005.05.018.

Great point. We cite this study in the revised manuscript and hope that we have made this point clear.

3. Lines 269-271 – Was it experimentally verified that the ectopic copy of DBP2 was expressed at normal levels?

Yes, we include these data in the revised manuscript in Figure 2—figure supplement 2D.

4. For all of the growth curves shown in Figure 2 and its supplement, we would be better able to assess the growth if the Y axis was a log scale since the cells are presumably growing exponentially. In addition, it's possible that the cell size or clumpiness of the mutants is different compared to wild type in a way that might affect the OD600 reading.

We have now included quantification of doubling time for all growth experiments throughout the manuscript for clarity. We sonicate yeast prior to OD measurements and culture seeding, which reduces clumping. Cell size differences also do not seem to be an issue in this case.

5. Figure 2 and supplement – The authors present convincing data that the mutant phenotypes observed in their dbp2 knockouts are not complemented by the integration of an exogenous copy of DBP2. However, in some sense, this is negative data, the failure to rescue the mutant phenotype. There are some additional strains whose analysis would be informative. Most obviously, if the dbp1 deletion phenotypes are really caused by reduced expression of MRP51, then integration of a second copy of MRP51 would be very strongly predicted to rescue the dpb1 deletion mutant phenotypes. Second, one would expect that the dpb1 and mrp51 deletions would have similar phenotypes, depending on how much residual Mrp51 remains in the dpb1 deletion.

Thank you for these suggestions. In the revised manuscript, we include rescue experiments showing rescue of all observed *dbp1∆::kanMX6* phenotypes by an exogenous copy of *MRP51* as suggested (Figure 2B-C; Figure 2—figure supplement 2A-C). As far as the second experiment, we are fortunate this was already done by other labs and suggests exactly what the reviewer predicts. Strains completely lacking *mrp51* cannot survive under conditions requiring respiratory growth (Stenger et al., *Microbial* Cell 2020) so the poor respiratory growth seen in our study is consistent with a hypomorphic phenotype based on the partial decrease in Mrp51 levels resulting from translational downregulation of this gene.

6. For the experiment in Figure 2F, were opposite orientations examined for these cassettes? From what is written in lines 400-402 it sounds like integration of the standard resistance cassettes in the opposite orientation would abolish the off-target effect in at least some of these cases. Is that known or predicted?

We agree that this is an interesting experiment and have included it in the revised manuscript (Figure 3G, Figure 3—figure supplement 2A). This indeed removes the respiratory growth defect that results from Mrp51 down-regulation and helps solidify our model.

7. line 461 -"Ablate" implies complete loss of function, which isn't the case for the examples shown. Use of a different word, such as "reduce" or "impair" would be more accurate. The severity of the effect would be better understood by comparison of the dbp1 and mrp51 deletions.

We have incorporated this suggested edit in the revised manuscript.

8. In the text the authors give their reconfigured cassette a simple and clear name – KANMX-ins. Therefore, for Figure 4 and its supplement, it would be clearer if the "KANMX-ins" name was used for the labels instead of the more complex name that is used.

Thank you for the suggestion, we have implemented the shorter name for these constructs and the newly added insulated plasmids into the relevant figures.

The comments above should be addressed by experimental or writing changes. The key experiment that should be included pertains to comment 5 above. That will test whether the integration of an ectopic copy of MRP51 rescues the dpb1 deletion phenotype and whether the dpb1 deletion phenotype mimics that of an mrp51 deletion. Both of these would greatly fortify the conclusion that the dpb1 phenotype is caused by reduced mrp51 expression.

Thank you for these excellent suggestions, we believe that we were able to address all comments, including point 5 above and hope that the reviewer finds the revised manuscript to be suitable for publication at *eLife.*

Reviewer #2 (Recommendations for the authors):Suggestions for the authors for the presentation of their study.General: if possible, please make clear at the level of individual figures and legends that all presented observations are reproducible. Examples would be if growth curves are n=2, potentially show both or show average and range, and state in figure legend instead of methods; for western blot quantification please indicate what the quantification represents – a single representative of experiment done more than once or an average of multiple experiments? If more than one, please add an error or range to the values. If one but representative, please state explicitly.

This has been done throughout the revised manuscript. For growth curve data we chose to calculate doubling time for each strain so that readers can easily compare mutant growth rates directly and incorporated biological replicate data into these plots. For each growth experiment we now show one representative curve in the main figure and provide doubling time for all biological replicates in the supplemental figure panels listed.

1. Line 191. "Together these data indicate that cassette replacement of the DBP1 ORF drives aberrant transcription independent of strain, cassette selection identity, and experimental conditions. Instead, the observed misregulation of MRP51 appears to be a general side-effect of the cassette-insertion."This statement is too strong. In the experiment shown, two marker cassettes that may or may not be related are tested, apparently in the identical orientation. It is not clear if hygMX4 is the standard name for this marker. Is what is meant hphMX4? hphMX4 and kanMX6 will both use TEF promoter and terminator and in the case here, it appears that both markers were used in the same orientation, which as we learn later is likely critical for the mechanism, though not explicitly tested. This means that conclusion that the effect is general for cassette insertion has not been demonstrated, and potentially would not even be expected (based on the understood mechanism).

Thank you for the suggestions, we have moved this statement to after the data presented in Figure 2D (previously presented in Figure 2F) and attempted to clarify the language to indicate that the effect is general to multiple cassettes though as the reviewer indicates is specific (not general) to cassette orientation. Additionally, we now have placed diagrams showing cassette makeup at all relevant figures such that redundant or unique sequences between cassettes can be easily determined.

We added the reviewer-suggested experiment of a flipped cassette inserted at the *DBP1* locus (Figure 3G), which demonstrates as the reviewer predicts, the off-target effect is specific to cassette orientation as predicted by our model. This new experiment helps solidify our proposed mechanism, whereby bidirectional promoter activity stemming from cassette promoters drives mis-regulation of neighboring genes.

We have also corrected the hygromycin cassette nomenclature and appreciate this note.

2. In Several places "believe" is used. Consider "propose" "argue" "hypothesize" "favor" in place.

Thank you for this suggestion, we have replaced these instances in the revised manuscript.

3. line 296 "Insertion of every resistance cassette tested lead to a similar growth defect in a non-fermentable carbon source (Figure 2F), a phenotype which was not dependent on Dbp1 (Figure 2B)."In the manuscript, as currently written, the concept of bidirectional transcription directed from the marker promoter is left to mention later. Since the tested markers share attributes, the bidirectional model is an attractive explanation at this position in the manuscript and a straightforward test of the proposed mechanism would be to use the markers in the opposite orientation. This experiment was not done but could be done. It has the added value of directly demonstrating that the effect is not innate to the marker but to the specific context in use.

We appreciate this suggestion and have included this experiment into the revised manuscript (Figure 3G), and as mentioned above believe it helps solidify our model. We have also restructured the manuscript, including swapping Figures 3 and 4, to bring this point up earlier and revised the manuscript title and abstract to emphasize the importance of bidirectional transcription earlier.

4. For figure 2A, an additional panel with a difference map of some kind between the two strains could have value in visually demonstrating that the selected groups of genes are or are not the only major groups affected. There appear to be other clusters also affected and a description of these could add value to the presentation – for example, if it were discussed what is known about cellular phenotypes when mitochondrial translation is perturbed in a manner similar to reduced expression of components and how these data merely recapitulate previous observations or go beyond to describe impacts on the proteome that have not been described.

This is an interesting point. The two clusters that we highlight are the ones for which the changes made sense as direct effects of the genetic change that we engineered (ie. it was possible to make a model for the changes based on known functions of either Dbp1 or Mrp51) but we agree with the reviewer that other clusters are interesting, as well. For example, there is an interesting cluster midway down the tree that shows high levels in lower fractions in the wild-type cells and is extremely highly enriched for the mitochondrial ATP synthase complex (suggesting that this is not highly assembled in the cells with low Mrp51 levels) and a group of proteins that are up in the *dbp1∆::kanMX6* cells (just above the 54S cluster), which includes proteins that transport proteins into the mitochondria (Tim54, Tom70, Sdh3) and not much else.

The type of data that we collected is unusual in that it is mass spectrometry of polysome fractions specifically, and thus there is not much comparable data in the literature. We think it may indeed be providing new insight into the cellular response to mitochondrial dysfunction.

5. For figure 2f, the lack of complementation is a negative result because it is not demonstrated that DBP1 at the ectopic locus is transcribed and translated to the same extent as at the native locus. This is a minor concern because of the use of insulated KO cassette later but examination of the described CRISPR allele, or reversed kanMX6 or hphMX4 cassettes, or a CRISPR allele removing the DPB1 promoter and TSS, or introduction of a stop codon early in the ORF would have been a nice addition to this experiment.

Yes, this is a great point. We have confirmed that Dbp1 is expressed to the same extent from the ectopic and endogenous locus using western blots and added these quantifications to the revised manuscript (Figure 2—figure supplement 2D). We have also bolstered these experiments by demonstrating that while ectopic Dbp1 expression does not rescue the *dbp1∆::kanMX6* phenotypes, ectopic expression of Mrp51 does (Figure 2B-C and Figure 2—figure supplement 2A-C). We appreciate these suggestions and believe this data improve the flow of the revised manuscript.

6. Figure 2- fs 1A. The WT sample presumably has an error in the measurement. Even though mutant and WT are normalized there should be a way to propagate the error to the WT to indicate it is 100% +/- SD.

Yes, thank you for catching this error, we apologize with the manner by which these results were previously presented and have fixed this plot in the revised manuscript.

7. Line 357. "For the last case, in which UPF1 was replaced with the NATMX6 cassette, an abundant aberrant transcript was observed." While it is noted later, it would be appropriate here to note that upf1∆ itself may alter the stability of cryptic transcripts and this could be the reason for the especially strong transcript seen here.

Yes, we find this to be a very interesting point! We initially did mention it earlier in the text, but some found it distracting without additional data and this led to its position in the Discussion rather than the Results.

In the revision process, we performed further analyses of RNA decay mutants, which revealed a role for *RRP6*-dependent exosome in degrading most cassette-driven divergent transcripts (Figure 5). This degradation likely depends on NNS-dependent termination pathways, which are responsible for terminating many endogenous divergent transcripts. We suggest that the ability of naturally occurring NNS termination sequences to prevent widespread transcription interference from cassette-driven divergent transcription may explain why some loci are susceptible to severe neighboring gene disruption upon cassette insertion and some are not.

It is interesting that cells lacking *RRP6* show even higher divergent transcript levels when a selection cassette is inserted at the *UPF1* locus than is seen with cassette-replacement of *UPF1* alone (Figure 5D). This may indicate that both NMD and exosome-mediated decay can target divergent cassette-driven transcripts in some cases. Given the lack of control here, however (ie. no case in which a cassette is inserted at the *UPF1* locus, but Upf1 expression is normal), the data for this locus is inconclusive. Moreover, we were unable to find additional data to support a role for NMD in degrading cassette-driven transcripts among published datasets, so discuss a possible role for NMD in the revised manuscript only briefly, when the data for *UPF1* cassette insertion are first introduced.

8. Line 370 "Seamless deletion of the DBP1 ORF by this strategy led to the production of a transcript containing only the 5' and 3' UTRs of DBP1. Expression of this mutant transcript led to the aberrant translation of a dubious ORF contained within the DBP1 3' UTR, not translated in wild type (Figure 3—figure supplement 1).It is a useful point to make that any genetic alteration may have unintended consequences (and this is a nice additional example) but the wording seems to unnecessarily conflate CRISPR-mediated editing with one specific time of deletion (removal of ORF) instead of the many different ways gene function may be rendered null using CRISPR editing. The statement just above states "effects" on "surrounding genes" plural but is not clear outside of the 3 UTR ORF as MRP51 shows a 1.2X effect- is this latter effect significant? If not, what other genes are affected? It does not seem necessary to use this example as justification for the new generation of selectable markers that are introduced in the next section- their value stands on their own, regardless of the extra effects of particular types of CRISPR editing.

We attempted to clarify this in the revised manuscript, as we agree this is not needed to justify new cassettes. The major point, which we hope we have improved our presentation of, is that small genomic changes can have unintended consequences on translation of coding sequences produced from neighboring sequences, a topic that is now in the discussion. We realize that the fold-change labels on this figure were unclear in their significance. We had labeled *MRP51* to demonstrate that it is relatively unchanged in the Cas9 unmarked *dbp1∆*, but realize this was unclear without further context.

Based on the reviewer’s feedback, we realized that there was a better organization of the data that we have implemented in the revised manuscript and that we think address several of the reviewer’s concerns. We moved the data for CRISPR-edited cells to Figure 5—figure supplement 2. We swapped the former Figures 3 and 4 and included new data for the reversed cassette insertion to the new Figure 3. This shows that bidirectional promoter activity from the cassette is driving the off-target effects. Finally, the new Figure 4 and Figure 5 now shows that this cassette-driven off-target effect occurs more broadly but masked by exosome-mediated degradation in many cases. We believe this organization solves several problems, including making the source of this off-target effect clear earlier in the manuscript.

9. Figure 3. An alternative framing would be that selectable markers employing the TEF1 promoter can drive or induce transcription divergently from the cassette in multiple contexts.

Yes, good point. With the manuscript reorganization, we believe that it is now more clearly stated that *pTEF-*based cassette insertion can drive divergent transcripts in multiple contexts. In fact, the *rrp6* data that are in Figure 5 now suggest that they always do so.

10. Line 492 "These features included a gene antisense and 5' to the cassette.."This could be more accurately described as KO cassettes can have two orientations relative to the knocked out genes and adjacent genes are not technically antisense unless they occupy the same sequence on the other strand. Consider "These features include an upstream gene transcribed divergently from that of the orientation of the selectable marker gene in the cassette".

It's true that divergent is the correct term to use here and we have fixed this wording to be clearer. We appreciate this comment.

11. One consideration that would greatly improve the value of the new selectable markers created would be a design strategy that diversifies insulator sequences such that cassettes would not share flanking sequences. One issue with the kanMX6/hphMX4/natMX etc cassettes is that they share TEF promoter and terminator. This means that using them in the same strain reduces the efficiency of use by about 10-fold, due to marker exchange instead of integration into the desired target when one cassette is already in use in a strain. Having additional same sequences on each end can exacerbate this. One possibility would be to switch 5' and 3' positions but another would be to deploy at least two additional terminator regions to reduce marker exchange propensity.

We appreciate this suggestion and fully agree. We have made two adjustments to reduce sequence overlap and created 2 additional insulated plasmids. First, we reasoned that the 5’ insulator flanking the cassette promoter is necessary to prevent the promoter driven off-target effects while the 3’ flanking insulator is dispensable because it sits adjacent to the cassette terminator. This was confirmed by swapping the orientation of the kanMX6 cassette integrated into the *DBP1* locus which places the *TEF* terminator adjacent to *MRP51* (Figure 3G, Figure 3—figure supplement 2A). Therefore, we removed the 3’ terminator and created a 5’ only *tDEG1* insulated version of the kanMX6-ins1 plasmid (creating kanMX6-ins6, Figure 3H). Next, we cloned another 5’ insulated cassette (kanMX6-ins7, Figure 3H) that only shares the primer amplification sites with kanMX6-ins6. This plasmid contains the *tCYC1* terminator 5’ of the *C. glabrata PGK1* promoter expressing *nat^r^,* and is terminated with the *C. glabrata PGK1* terminator. To confirm the efficacy of these new insulated cassettes we integrated them into the *DBP1* locus exactly as done with the prior cassettes and tested whether the resulting strain exhibited a growth defect in non-fermentable carbon sources. Integration of both kanMX6-ins6 and kanMX6-ins7 did not induce growth defects in the non-fermentable carbon source glycerol when used to replace *DBP1,* signifying their lack of disruption of *MRP51* (Figure 3-supplement 2B). We will deposit these improved insulated plasmids, as well as our original proposed insulated cassettes, at Addgene for easy access.

12. Line 608 "c terminally" → "C-terminally" or "carboxy-terminally".

Thank you for this correction, we have updated the text in the revised manuscript.

13. Line 634 "100 µM" of what?

Cycloheximide, thank you for noting this error, we noted this in the revised manuscript and appreciate your keen eye.

14. Line 652 Fischer → Fisher.

Thank you, we have now corrected this typo.

15. Line 751 "We then cloned 3 additional cassettes using varying selection (including Saccharomyce *S. pombe* HIS5: complements S.cerevisiae HIS3, TRP1, and NATr)…"Please make clear in methods if these markers were used to replace kan in kanMX6 or if insulators were added to those original cassettes and cite the source of the original cassettes (plasmids) and their official cassette names here for the relevant markers.

These additional selection markers were used to replace kan^r^ in the kanMX6-ins2 plasmid and their description is listed in supplemental table 1 and schematized in Figure 3. We have more clearly listed the plasmid that each resistance marker was amplified from and noted in the text that each of these new cassettes all share the same flanking sequence as that in the kanMX6-ins2 plasmid. We have also cloned an additional insulated cassette with non-shared flanking sequences that uses the promoter and terminator from *C. glabrata PGK1* to express the nat^r^ gene, to further facilitate multiple gene replacements within the same strain and reduce the likelihood of marker exchange. We also provided diagrams clearly showing cassette makeup to clarify shared or unique sequences between cassettes for these new selection cassettes (please see suggestion 11 for more information).

16. Line 756. 5' RACE. The nature of the sequencing here is not described. Numbers of "reads" are described, but is it instead meant numbers of clones with cDNA ends mapping to indicated positions because that is my understanding of how the GeneRacer III TOPO kit is designed to be employed. Additionally, were RACE products attempted to be mapped within the cassette and not found, or was it not examined? Are the new products at MRP51 within or external to the cassette and how close to the junction? It would be useful to know the distance between the cassette TSSs for the resistance marker and the induced divergent transcription to get an idea of if the properties are consistent with divergent transcription as described for single nucleosome-free regions or if instead the TEF one sequence used in these cassettes actually has an additional promoter region attached

Yes, the reviewer is correct, here the number of “reads” listed signifies the number of cDNA clones whose 5’ end of transcript mapped to that position. We have changed the language in the text to more clearly represent this and have added the information requested. None of the 5’ RACE products sequenced mapped to the cassette and all clones analyzed are listed and depicted in this figure, apart from a single clone whose 5’ end of transcript mapped within the *MRP51* ORF and likely represented a fragmented transcript not a true 5’ cDNA end. The finding that these cassette driven transcripts do not contain any cassette sequence is consistent with our previous observations from the mRNA sequencing data in Figure 1 and Figure 1—figure supplement 1 where the 5’ extended transcripts for *MRP51* did not include any cassette sequence. Instead, the TSS for the divergent cassette driven transcript is ~530bp away from the TSS of the sense transcript that drives the expression of kan^r^, and ~110bp away from the cassette-genome junction. Interestingly, the distance between the coding and divergent transcript produced from the kanMX6 *TEF* (*A. gossypii)* promoter appears to be highly similar to that observed at the *S. cerevisiae TEF1* locus (Figure 5E; Xu et al., *Nature* 2009).

Because of this similarity, and the fact that we have sequenced the plasmids and confirmed there are no additional promoters, we believe the divergent transcript originates from the same nucleosome free region as the sense transcript.

Furthermore, we see the divergent transcript-driven phenotypes resulting from Mrp51 disruption with insertion of the *TRP1* cassette, which shares only the 5’ and 3’ flanking primer amplification sequences (Figure 2D). However, this common primer sequence is still present in the insulated cassettes (and not itself insulated) so it cannot be responsible.

We should have noted (and do so in the revised manuscript), that *pTEF1* has bidirectional promoter activity along with most promoters, but the resultant transcript from the endogenous locus is subject to rapid exosome-based degradation and thus not visible in WT mRNA-seq data (Xu et al. *Nature* 2009).

References:

Almada, A. E., Wu, X., Kriz, A. J., Burge, C. B. & Sharp, P. A. Promoter directionality is controlled by U1 snRNP and polyadenylation signals. *Nature* 499, 360–363 (2013).

Ben-Shitrit, T. *et al.* Systematic identification of gene annotation errors in the widely used yeast mutation collections. *Nature Methods* 9, 373–378 (2012).

Carroll Kristina L., Pradhan Dennis A., Granek Josh A., Clarke Neil D., & Corden Jeffry L. Identification of cis Elements Directing Termination of Yeast Nonpolyadenylated snoRNA Transcripts. *Molecular and Cellular Biology* 24, 6241–6252 (2004).

Chen J. et al. Kinetochore inactivation by expression of a repressive mRNA. *eLife* 6, 10.7554/*eLife*.27417 (2017)

Cheng Z. and Otto G.M. et al., Pervasive, coordinated protein level changes driven by transcript isoform switching during meiosis. *Cell* 172, 5:910-923 (2018)

Chia M. et al. Transcription of a 5' extended mRNA isoform directs dynamic chromatin changes and interference of a downstream promoter. *eLife* 6, 10.7554/*eLife*.27420 (2017)

Churchman, L. S. & Weissman, J. S. Nascent transcript sequencing visualizes transcription at nucleotide resolution. *Nature* 469, 368–373 (2011).

Core, L. J., Waterfall, J. J. & Lis, J. T. Nascent RNA Sequencing Reveals Widespread Pausing and Divergent Initiation at Human Promoters. *Science* 322, 1845–1848 (2008).

Costanzo M. et al. The Genetic Landscape of a Cell. *Science* 327, 5964:425-431 (2010)

Curtin J.A. et al. Bidirectional promoter interference between two widely used internal heterologous promoters in a late-generation lentiviral construct. *Gene Therapy* 15, 5:384-390 (2008)

Egorov, A. A. *et al.* A standard knockout procedure alters expression of adjacent loci at the translational level. *Nucleic Acids Research* 49, 11134–11144 (2021).

Floor S.N. & Doudna J.A. Tunable protein synthesis by transcript isoforms in human cells. *eLife* 5, 10.7554/*eLife*.10921 (2016)

Gherman A., Wang R., Avramopoulos D. Orientation, distance, regulation and function of neighbouring genes. *Human Genomics* 3, 2:143 (2009)

Gidoni D. et al. Bidirectional SV40 Transcription Mediated by Tandem Sp1 Binding Interactions. *Science* 230, 4725:511-517 (1985)

Hollerer I. et al. Evidence for an Integrated Gene Repression Mechanism based on mRNA Isoform Switching in Human Cells. *G3* 9, 4:1045-1053 (2019)

Neil, H. *et al.* Widespread bidirectional promoters are the major source of cryptic transcripts in yeast. *Nature* 457, 1038–1042 (2009).

Ntini E. et al. Polyadenylation site–induced decay of upstream transcripts enforces promoter directionality. *Nature Struc & Molec Bio* 20, 8:923-928 (2013)

Preker, P. *et al.* RNA Exosome Depletion Reveals Transcription Upstream of Active Human Promoters. *Science* 322, 1851–1854 (2008).

Schulz, D. *et al.* Transcriptome Surveillance by Selective Termination of Noncoding RNA Synthesis. *Cell* 155, 1075–1087 (2013).

Seila, A. C. *et al.* Divergent Transcription from Active Promoters. *Science* 322, 1849–1851 (2008).

Stenger M. et al. Systematic analysis of nuclear gene function in respiratory growth and expression of the mitochondrial genome in *S. cerevisiae*. *Microbial Cell* 7, 9:234-249 (2020)

Teodorovic, S., Walls, C. D. & Elmendorf, H. G. Bidirectional transcription is an inherent feature of Giardia lamblia promoters and contributes to an abundance of sterile antisense transcripts throughout the genome. *Nucleic Acids Res* 35, 2544–2553 (2007).

Trinklein N. D. et al. An Abundance of Bidirectional Promoters in the Human Genome. *Genome Research* 14, 1:62-66 (2004)

Van Dalfsen K.M. et al. Global Proteome Remodeling during ER Stress Involves Hac1-Driven Expression of Long Undecoded Transcript Isoforms. *Dev Cell* 46, 2:219-235 (2018)

Vander Wende H.M. Meiotic resetting of the cellular Sod1 pool is driven by protein aggregation, degradation, and transient LUTI-mediated repression. *bioRxiv* https://doi.org/10.1101/2022.06.28.498006 (2022)

Xu Z. et al. Bidirectional promoters generate pervasive transcription in yeast. *Nature* 457, 7232:1033-1037 (2009)